# Influence of anthropogenic pollution on the molecular composition of organic aerosols over a forest site in the Qinling Mountains region of central China

Xin Zhang[1,2], Lijuan Li[1], Jianjun Li[1], Yue Lin[1], Yan Cheng[2], Rui Wang[1], Shuyan Xing[1], Chongshu Zhu[1], Junji Cao[1,3,*], and Yuemei Han[1,4,*]

[1]Key Laboratory of Aerosol Chemistry and Physics, State Key Laboratory of Loess Science, Institute of Earth Environment, Chinese Academy of Sciences, Xi'an 710061, China

[2]School of Human Settlements and Civil Engineering, Xi'an Jiaotong University, Xi'an 710049, China

[3]Institute of Atmospheric Physics, Chinese Academy of Sciences, Beijing 100029, China

[4]National Observation and Research Station of Regional Ecological Environment Change and Comprehensive Management in the Guanzhong Plain, Xi'an 710061, China

*Correspondence to*: Yuemei Han (yuemei.han@ieecas.cn), Junji Cao (jjcao@mail.iap.ac.cn)

**Abstract.** Biogenic organic aerosols interacting with anthropogenic pollutants lead to large uncertainties in aerosol properties and impacts, yet the underlying mechanisms remain to be fully elucidated. To explore the anthropogenic–biogenic interactions in the Qinling Mountains region of central China, we investigated the molecular composition of organic aerosols in atmospheric $PM_{2.5}$ at a forest site in summer and winter of 2021/2022, using ultrahigh performance liquid chromatography coupled with Orbitrap mass spectrometry. Organic species were more abundant and chemically diverse in winter compared with those in summer, as revealed by their higher numbers and peak area intensities. The molecular characteristics of organic species exhibited distinct seasonal variabilities, with higher peak-area-weighted mean values of molecular weight and oxidation state but lower unsaturation degree in summer, possibly associated with more biogenic emissions and intense photochemical processes. A variety of organic tracer species were identified in the two seasons, among which the biogenic ones were relatively more abundant in summer, contrasting with the substantial increase of anthropogenic ones in winter. A higher ambient relative humidity, except for heavy precipitation, usually promoted the production of nitrogen- and sulfur-containing organic species by involving more anthropogenic pollutants. The synergistic effects of meteorology and anthropogenic pollution greatly affected the organic aerosol production in this forest atmosphere, thereby altering their molecular composition and related properties under different environmental conditions. The combined set of results herein provides direct evidence for the anthropogenic perturbations on air quality, atmospheric chemistry, and associated climate impacts in the Qinling Mountains region.

## 1 Introduction

Atmospheric aerosol particles generated from various natural and anthropogenic sources have substantial impacts on Earth's climate, regional air quality, and human health (Liu et al., 2019; McNeill, 2017). Biogenic organic aerosols comprise an important component of aerosol particles globally (Mahilang et al., 2021; Shrivastava et al., 2017). Terrestrial vegetation emits huge amounts of volatile organic species and primary biological particles into the atmosphere, which constitute the main sources of biogenic organic aerosols (Després et al., 2012; Rap et al., 2018; Wang et al., 2024a). A number of laboratory and field studies have demonstrated that the presence of anthropogenic pollutants (e.g., $NO_x$, $SO_2$, $NH_3$, sulfate, and etc.) may alter the physicochemical properties and production yields of biogenic organic aerosols (Carlton et al., 2018; Hoyle et al., 2011; Zhao et al., 2016). However, the interactions of anthropogenic pollution with biogenic organic aerosols vary considerably across spatial and temporal scales, which lead to large uncertainties in assessing their environmental and climate related

impacts (Dong et al., 2022; Matsui et al., 2014; Xu et al., 2021a). Therefore, despite the enormous efforts of existing studies, the underlying mechanisms of anthropogenic–biogenic interactions remain to be fully elucidated.

Anthropogenic pollution can affect biogenic organic aerosol through various atmospheric chemical and physical processes, as previously proposed in the literature. They could promote the oxidation of biogenic volatile species for more organic aerosol production by increasing the levels of atmospheric oxidants (e.g., hydroxyl, ozone, and nitrate radicals) (Ng et al., 2017; Wu et al., 2020; Xu et al., 2015). The reaction chemistry of biogenic precursors would be changed under different $NO_x$ levels, leading to complex chemical composition and yields of the secondary products (Han et al., 2016; Xu et al., 2014; Yan et al.,

2020; Zhao et al., 2018). Meanwhile, anthropogenic particulate matter provides important surface for gas–particle partitioning and particle-phase reactions of biogenic organic products, whereas these processes were to a large extent governed by the particle properties such as acidity and physical state (Pye et al., 2020; Reid et al., 2018). In addition, acidic pollutants (e.g., $SO_2$, $NO_x$, and sulfate) may involve in formation of organosulfur and organonitrogen compounds by reacting with biogenic aerosol in various ambient environments (Brüggemann et al., 2020; Fan et al., 2022). Nevertheless, previous relevant studies

are primarily focused on the characterization of bulk organic matter or fragments using traditional techniques such as online aerosol mass spectrometry (Noziere et al., 2015; Wang et al., 2021a). A thorough investigation of anthropogenic–biogenic interactions at the molecular level is crucial to gain deep insights into the mechanisms and impacts.

The Qinling Mountains region is primarily situated in the southern part of Shaanxi Province in central China (Fig. 1). It serves as an important natural geographical and climatic boundary between the northern and southern China (Yang et al., 2015; Yao

et al., 2020). This region also lies on the edge of the Asian monsoon region and is quite sensitive to climate change (Cai et al., 2010). The unique geographical features of this region have attracted great interests for ecological and environmental research (Wang et al., 2021b; Zhang et al., 2022). The Qinling Mountains are enriched in a huge variety of forested vegetation (Wang et al., 2021c; Zhao et al., 2014), such as *Quercus*, *Betula*, conifer, and subalpine meadows, contributing large amounts of biogenic primary emissions and secondary oxidation products into the atmosphere (Cao et al., 2022; Li and Xie, 2014; Xu et

al., 2020). Also, this region is often affected by anthropogenic air pollutants from the surrounding rural and urban areas in the Guanzhong Plain of northwest China (Cao and Cui, 2021; Zhao et al., 2015). Previous studies on atmospheric aerosols in this region were primarily focused on the remote environments at high altitudes, such as Mountains Hua and Taibai (Li et al., 2013; Meng et al., 2014; Shen et al., 2023; Li et al., 2023). From those work, the characteristics of inorganic species and the formation and evolution processes of organic aerosols are gradually being understood (Li et al., 2020; Niu et al., 2016). However, the

interactions of anthropogenic pollution with biogenic organic aerosols and the potential impacts are still rarely reported at the lower altitudes of this region to date, especially at the anthropogenic–biogenic intersection zones. Understand these interactions will be valuable to elucidate the anthropogenic perturbations on air quality, atmospheric chemistry, and associated climate impacts in the Qinling Mountains region.

The present study investigates the influence of anthropogenic pollution on molecular characteristics of atmospheric organic

aerosols in the Qinling Mountains region, in order to advance our current knowledge on the complex interactions between anthropogenic pollutants and biogenic emissions. The molecular composition of organic aerosols in ambient $PM_{2.5}$ was characterized at a forest site in the northern foothills of this region during contrasting summer and winter seasons of 2021–2022, using ultrahigh-performance liquid chromatography coupled with electrospray ionization and Orbitrap mass spectrometry. The chemical compositional and structural variability of organic aerosols were presented and discussed

thoroughly to understand the underlying causes across the two seasons. The anthropogenic and biogenic contributions to organic molecular composition were further examined based on a variety of organic tracer species. Finally, the effects of meteorological parameters and anthropogenic pollutants were explored to gain insights into the potential pathways of organic aerosol production in this forest atmosphere. So far to our knowledge, this study for the first time reports the overall molecular characteristics of organic aerosols between contrasting seasons and provides direct evidence for the large influence of

anthropogenic pollution in the Qinling Mountains region based on high-resolution Orbitrap mass spectrometry.

## 2 Methodology

### 2.1 Site and aerosol sampling

Atmospheric $PM_{2.5}$ was sampled at a forest site (34.06° N, 108.34° E, around 530 m above sea level) in the Qinling Mountains region of central China during summer and winter seasons of 2021/2022. This site is situated in the northern foothill of the Qinling Mountains and approximately 50 km southwest of the megacity Xi'an, as shown in Fig. 1. It belongs to the National Observation and Research Station of Regional Ecological Environment Change and Comprehensive Management in the Guanzhong Plain. Since anthropogenic pollutants and biogenic emissions were prevalent in this area, it is thus considered as an anthropogenic–biogenic intersection zone. A medium-flow air sampler (HC-1010, Qindao Hecheng Ltd., China) was operated at 100 L $min^{-1}$ for collecting $PM_{2.5}$ samples on precombusted quartz fiber filters (90 mm in diameter, Whatman Inc., USA). The sampling height was approximately 3.5 m above the ground. Each sample collection lasted around 23 h from 10:00 am to 9:00 am of the next day in local time. A total of 33 aerosol samples were collected during the entire study period, including 17 and 16 samples in summer and winter, respectively. Four field blanks were also collected using the same approach but with the sampler pump turned off. All the samples were stored in a freezer at −20 °C for further chemical analysis.

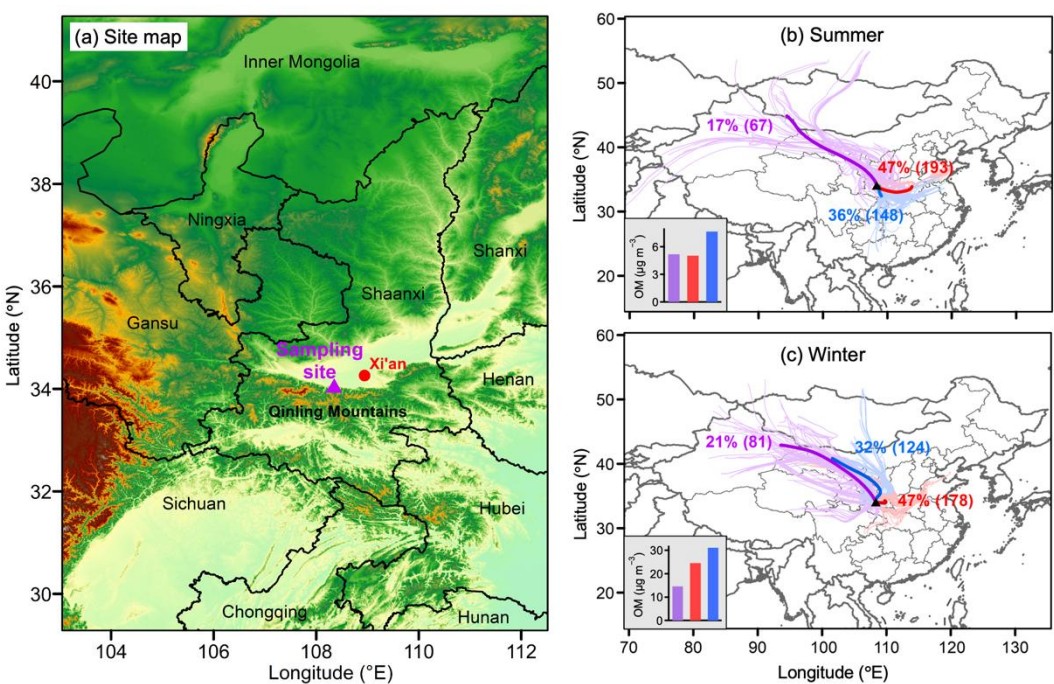

**Figure 1. (a) A map showing the sampling site at northern foothill of the Qinling Mountains region in central China. (b, c) Back trajectories of air masses arrived at 500 m above the ground level over the sampling site and their cluster analysis (represented by the thick solid lines with percentages and numbers) during the summer and winter periods. The bottom left panels in (b, c) present the mean mass concentrations of organic matter that were measured using the carbon analyzer for aerosol samples collected on the days with the corresponding air mass directions.**

### 2.2 UHPLC–HRMS measurement and data processing

A quarter of each sample in 12.56 $cm^2$ area was ultrasonically extracted using 9 g acetonitrile and water mixture in 9:1 volume for 30 min (3 g for 10 min, repeated three times). The extraction system was placed in a water–ice bath to eliminate potential evaporation or chemical reactions of aerosol components. The extracts were filtered through 0.2 μm pore-size

polytetrafluoroethylene membranes (Pall Co., USA) and then concentrated to 500 μL under a gentle nitrogen stream. The organic molecular composition of individual samples was subsequently measured using a Vanquish Flex ultrahigh-performance liquid chromatograph coupled with a high-resolution Q-Exactive Orbitrap mass spectrometer (UHPLC–HRMS, Thermo Scientific Inc., Germany). For each sample, a volume of 5 μL analyte was injected and separated using a Thermo Hypersil Gold $C_{18}$ column (100 × 2.1 mm, 1.9 μm) with two mobile phases consisting of (A) 0.1% formic acid in ultrapure water and (B) acetonitrile. A gradient elution at a flow rate of 0.25 mL min$^{-1}$ was performed for 25 min, including initially 5% B at 0–2.5 min, linearly increased to 50% B at 2.5–8 min with a 3-min hold, linearly increased to 95% B at 11–17.5 min with a 5.5-min hold, and then dropped to 5% B within 0.2 min and held for 1.8 min. The analyte was then introduced into a heated electrospray ionization source (ESI) and determined by the HRMS. Raw data were acquired in both negative and positive ESI modes (i.e., ESI− and ESI+) at the spray voltages of −3 and 4 kV, respectively. The mass range was set at $m/z$ 50–750 in full MS scan mode, with the mass resolution of approximately 140,000 at $m/z$ 200. The HRMS was externally calibrated using Thermo Scientific Pierce standard calibration solutions before the measurement.

The raw data from UHPLC–HRMS analysis was processed using an open-source software MZmine 2.53 (Pluskal et al., 2010). The main procedures were similar to our previous work (Li et al., 2024a; Lin et al., 2022). Briefly, the elemental composition of organic molecular species was constrained to $C_{1-40}H_{1-80}O_{0-50}N_{0-4}S_{0-2}$ in the two ionization modes, using a mass tolerance of 2 ppm. An additional Na atom was also applied for possible sodium adducts in ESI+ mode. The assigned molecular species were further screened using the elemental ratios of H/C (0.3–3.0), O/C (0–3), N/C (0–1.3), and S/C (0–0.8) to ensure their presence in nature. The formulas disobeying the nitrogen rule for even electron ions were excluded for further analysis (Zielinski et al., 2018). The field blanks were also extracted and chemically analyzed using the same procedures, the results of which were used to correct any potential artifacts and backgrounds for the aerosol samples. The oxidation state of carbon atoms ($\overline{OS}_c$) in a molecule was calculated as $2 \times$ O/C − H/C for individual organic species (Kroll et al., 2011). The number of rings and double bonds in organic molecules was characterized with ring and double-bond equivalent (DBE), which was calculated as $(2 + 2c + n - h)/2$ by assuming the trivalent nitrogen and divalent sulfur atoms (Kind and Fiehn, 2007; Yassine et al., 2014). All the analyses and discussions in this study refer to the neutral organic molecules. Since the measurements of fragmentation patterns and reference standards were not available herein, the identification confidence of the presented results belongs to the unequivocal molecular formula level, according to Schymanski et al. (2014).

**2.3 Other analyses**

Each filter sample was weighted before and after sampling using an electronic microbalance under well-controlled temperature (20 ± 2 °C) and relative humidity (30 ± 2%) conditions, the difference of which dividing by the sampled air volume was used for calculating $PM_{2.5}$ mass concentration. The organic and elemental carbons on the filter samples were measured with the interagency monitoring of protected visual environments protocol using a thermal/optical carbon analyzer (model 2001A, Desert Research Institute, USA) (Chow and Watson, 2002). The mass concentration of organic matter was then estimated by 2.1 times that of organic carbon for this forest site (Turpin and Lim, 2001). Water-soluble inorganic ionic species on the samples were determined using a Metrohm ion chromatograph (model 940 Professional IC Vario) with routine procedures of our group (Zhang et al., 2011).

The hourly air quality data of $SO_2$, $NO_2$, CO, and $O_3$ during the study periods were obtained from the National Environmental Monitoring Centre of China (http://www.cnemc.cn) at the nearest monitoring station. The sulfur and nitrogen oxidation ratios (SOR and NOR) in the atmosphere were calculated respectively as $nSO_4^{2-}/(nSO_4^{2-} + nSO_2)$ and $nNO_3^-/(nNO_3^- + nNO_2)$, in which $n$ refers to the molar concentration (Kaneyasu et al., 1995). The odd oxygen ($O_x = NO_2 + O_3$) concentration was used to reflect atmospheric oxidation capacity (Herndon et al., 2008). The meteorological parameters of ambient temperature, relative humidity (RH), wind speed, and precipitation at three-hour intervals were downloaded from the Meteomanz database (http://www.meteomanz.com). In addition, the backward trajectories of air masses (500 m height above ground level and 72

h duration) arrived at the sampling site were calculated hourly across the study periods using the Hybrid Single Particle
Lagrangian Integrated Trajectory model (v5.2.1) (Stein et al., 2015).

## 3 Results and discussion

### 3.1 Seasonal variability in molecular composition of organic aerosols

Figure 2 presents the temporal variations of chemical components in PM$_{2.5}$, gas pollutants, and meteorological conditions during the entire study periods. The PM$_{2.5}$ mass concentration ranged at 12–56 (mean 29 ± 10) μg m$^{-3}$ in summer, which was much lower than those in winter (49–164, mean 89 ± 31 μg m$^{-3}$) (Table 1). Among the quantified chemical species, organic matter was the most dominant component that accounted for on average 22% of PM$_{2.5}$ mass in summer and 25% in winter (Fig. 2a, b). The inorganic salts of sulfate, nitrate, and ammonium were also abundant in both seasons. The mean mass fractions of sulfate were relatively higher in summer (11% vs. 8% in winter), whereas those of nitrate and ammonium were much higher in winter (14% and 7% vs. 4% of both in summer). This result suggests that the typical atmospheric conditions in summer (e.g., strong solar radiation and high temperature) would facilitate the chemical conversion of SO$_2$ into sulfate, while nitrate and ammonium were more efficiently produced under winter conditions. The PM$_{2.5}$ mass also consisted of 16–86% other components, including some additional ionic species, mineral dust, crustal materials, and etc. Overall, the seasonal variation tendencies of PM$_{2.5}$ and its chemical components at this forest site to some extent resembled those reported in the nearby rural and urban areas of Guanzhong Plain (Cao and Cui, 2021; Han et al., 2023; Zhao et al., 2015), where coal and biomass combustion and unfavorable atmospheric dispersion conditions were highly prevalent during winter heating seasons.

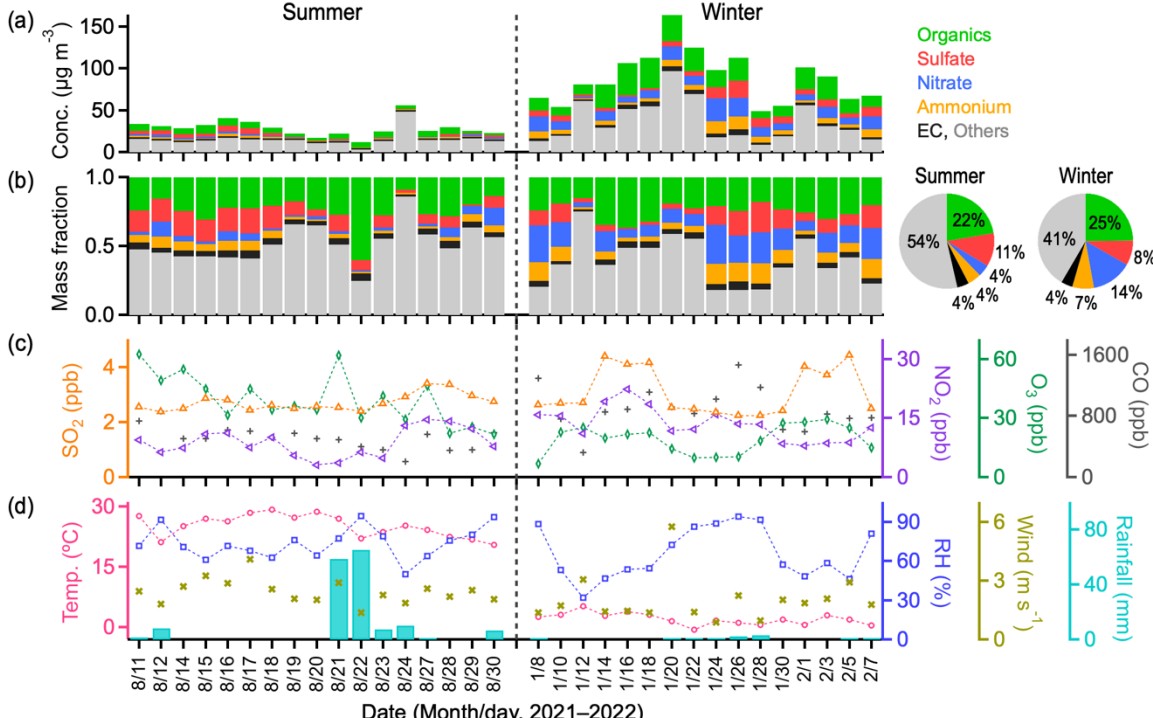

**Figure 2. Temporal variations of the (a) mass concentrations and (b) mass fractions of chemical components in ambient PM$_{2.5}$, (c) gas pollutant concentrations, and (d) meteorological factors over the studied periods in 2021/2022. The two pie charts in panel (b) represent the averaged mass fractions of PM$_{2.5}$ components in summer and winter periods.**

The distinct PM$_{2.5}$ mass concentration and chemical composition between summer and winter seasons over this forest

atmosphere can be attributed to multiple factors such as emission sources, formation pathways, gaseous precursors, meteorological conditions, and air mass origins. The gas pollutants of $SO_2$, $NO_2$, and CO had higher concentration levels in winter (mean 3.1, 14, and 887 ppb) than those in summer (2.7, 9, and 511 ppb) (Fig. 2c and Table 1), suggesting the large contributions of combustion sources such as fossil fuel and biomass burning (Mallik and Lal, 2014; Wei et al., 2023). In contrast, the higher $O_3$ levels in summer (mean 39 ppb) would facilitate atmospheric oxidation processes to produce more secondary aerosols compared with those in winter (19 ppb). Moreover, the studied summer period was characterized by the higher values of temperature, RH, wind speed, and more precipitation (Fig. 2d and Table 1). These typical ambient conditions in summer would promote the biogenic emissions (especially isoprene), aqueous phase chemistry, and disperse of air pollutants, respectively (Gu et al., 2023; Guo et al., 2016; Seco et al., 2022; Vettikkat et al., 2023). The air mass origins were also quite different in the two seasons (Fig. 1b, c). They primarily originated from the eastern (47%) and adjacent southern (36%) areas in summer, contrasting to those from the adjacent eastern (47%) and long-range northern (32%) areas in winter. The prevalent air masses with adjacent southern origins in summer at the studied site revealed the large biogenic emission contributions from the forested vegetation in the Qinling Mountains region. On the contrary, the long-range transported air masses from northwestern area (17% in summer and 21% in winter) were mostly clean ones with relatively lower levels of organic pollutants (see the small embedded panels in Fig. 1b, c).

**Table 1. The ranges and mean values of PM$_{2.5}$ chemical components, gas pollutants, and meteorological factors over the studied summer and winter periods in 2021/2022.**

|  | Summer | Winter |
|---|---|---|
| *Particulate mass concentration ($\mu g\ m^{-3}$)* | | |
| PM$_{2.5}$ | 12–56 (29 ± 10) | 49–164 (89 ± 31) |
| Organics | 3.1–9.9 (6.5 ± 1.9) | 8.8–38.7 (22.1 ± 9.5) |
| Sulfate | 0.1–7.0 (3.3 ± 2.2) | 1.3–20.1 (7.5 ± 4.7) |
| Nitrate | 0.2–3.3 (1.2 ± 1.0) | 2.8–28.0 (12.4 ± 6.3) |
| Ammonium | 0.1–2.8 (1.3 ± 0.9) | 1.2–15.4 (6.6 ± 4.0) |
| EC | 0.5–2.1 (1.2 ± 0.5) | 1.3–6.8 (3.6 ± 1.7) |
| Others | 3.1–48.2 (15.6 ± 9.0) | 9.2–96.7 (37.0 ± 24.9) |
| *Gas pollutants concentration (ppb)* | | |
| NO$_2$ | 3–15 (9 ± 4) | 8–22 (14 ± 4) |
| SO$_2$ | 2.4–3.4 (2.7 ± 0.3) | 2.2–4.4 (3.1 ± 0.9) |
| O$_3$ | 22–62 (39 ± 13) | 7–29 (19 ± 7) |
| CO | 204–855 (511 ± 154) | 319–1468 (887 ± 281) |
| *Meteorological factors* | | |
| Temp. (℃) | 20.4–29.2 (25.1 ± 2.8) | −0.6–5.2 (2.0 ± 1.5) |
| RH (%) | 50–94 (74 ± 12) | 32–94 (66 ± 20) |
| Rainfall (mm) | 0.2–65.4 (19.7 ± 26.4) | 0.2–3.2 (1.2 ± 1.1) |
| Wind (m s$^{-1}$) | 1.4–4.1 (2.4 ± 0.6) | 0.9–5.7 (2.0 ± 1.2) |

Figure 3 presents the molecular compositions of organic aerosols detected in ESI− and ESI+ modes during the summer and winter periods. The total numbers of organic molecular species assigned in ESI− mode were 879–2073 in summer and 1674–2873 in winter (Fig. 3a), corresponding to those of 1684–3188 and 2263–3648 in ESI+ mode (Fig. 3c). The total peak area intensities of organic species in both modes exhibited similar variation trends as those of species numbers (Fig. 3b, d), that is, with relatively higher values (approximately two times) observed in winter than those in summer. They were also consistent with the absolute mass concentrations of organic matter and PM$_{2.5}$ in the two seasons (Fig. 2a). These results together revealed

that organic aerosols in this forest atmosphere were more abundant and chemically diverse in winter compared with those in summer. In addition, both the number and peak area intensity of organic species were relatively higher in ESI+ mode, possibly resulting from the higher proton affinity and gas phase basicity of organic molecules that were readily ionized and detected than those in ESI− mode (Kiontke et al., 2016; Liigand et al., 2017).

The assigned organic species primarily consisted of those in CHO, CHON, CHOS, CHONS, and other oxygen-free groups

based on the elemental composition in the molecules. Most of these groups in the ESI− and ESI+ modes had greater species number and peak area intensity in winter than those in summer (Fig. 3a–d), which is consistent with the higher levels of air pollutants during the wintertime. In ESI− mode, the number fractions of organic species in both seasons were nearly comparable among the four primary groups of CHO, CHON, CHOS, and CHONS, ranging at 22–35%, 19–34%, 18–24%, and 15–28%, respectively (Fig. 3e). Nevertheless, the peak area intensity was predominated by the CHO species, followed by the

CHOS species (Fig. 3f). This result can be attributed to either the large abundance of these species in this forest atmosphere or their high ionization and transmission efficiencies under ESI− mode, despite that quantitative analyses of the highly complex composition of organic aerosols remain challenging due to the lack of authentic standards (Evans et al., 2024; Ma et al., 2022; Noziere et al., 2015). Moreover, the peak area fractions of CHON species increased on average 164% in winter compared with those in summer, which might be partly associated with the involvement of enhanced inorganic nitrogen species such as nitrate

and $NO_2$ during the wintertime (Fig. 2b, c). More CHON species could also favorably partition into the particle phase under the low ambient temperature conditions in winter (Bejan et al., 2007; Li et al., 2024b).

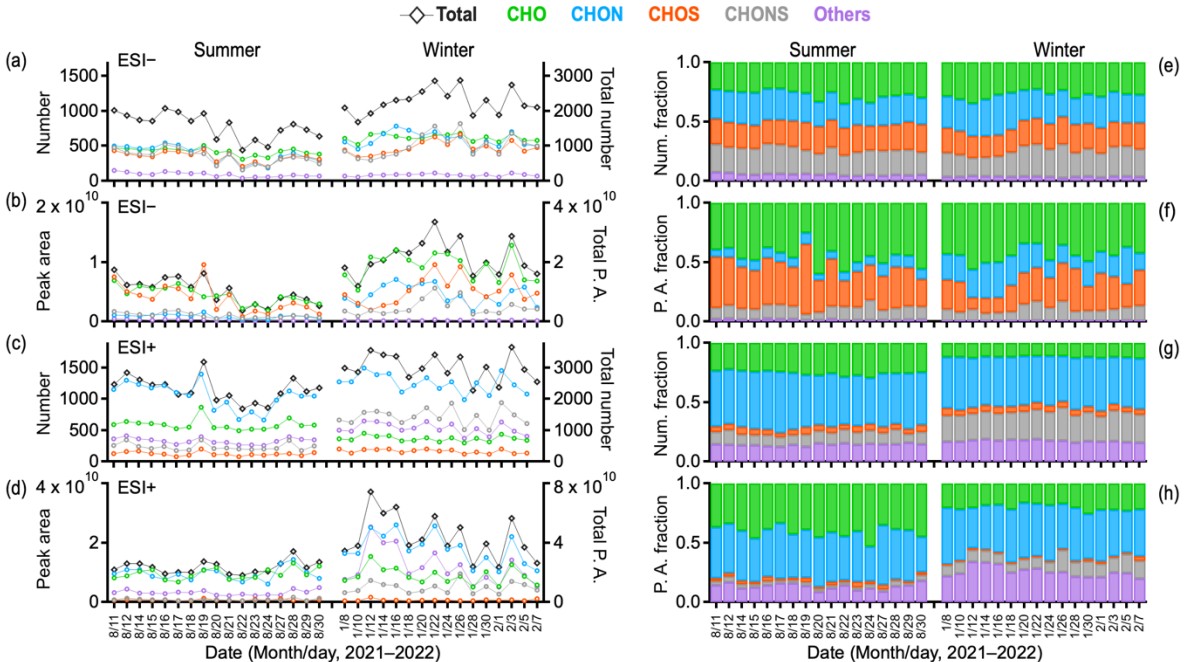

**Figure 3. (a, c) Species number and (b, d) peak area intensity of organic molecular composition (including each subgroup and total species), along with their corresponding number and peak area fractions (e–h), obtained from the UHPLC–HRMS analysis in negative and positive ESI modes over the study periods.**

In contrast, the molecular composition of organic species had different variation patterns in the ESI+ mode. The CHON species

were primarily dominant in terms of both number and peak area intensity, respectively accounting for 38−52% and 29−50% in those of total species across the entire period (Fig. 3g, h). Potential candidates for these CHON species included those with amino functional groups in the molecules (amines, amides, amino acids, and etc.), since protonated compounds were more sensitively detected in ESI+ mode. The CHOS species were very minor components in this mode, which can be primarily due

to their acidic nature (Lin et al., 2012a; Wang et al., 2017a). The number of CHONS species and the peak area intensity of other oxygen-free species (mainly CHN here) apparently increased in the winter period, while both the number and peak area intensity of CHO species inversely decreased. One proposed explanation is that the presence of high levels of anthropogenic air pollutants (e.g., $SO_2$, $NO_2$, sulfate, and nitrate; Fig. 2) during the wintertime facilitated the production of more chemically diverse organic species by reacting with organic precursors such as carbonyls, alcohols, and carboxyls (Fan et al., 2022; Xu et al., 2021b).

In addition to the observed seasonal variability of organic molecular composition, a number of organic species (286 in ESI− and 277 in ESI+ modes) were commonly found in both summer and winter samples (Fig. 4a, b), which could be originated from the same emission sources and/or reaction pathways. Nevertheless, these common species generally accounted for a small portion in the total number of assigned molecular formulas in individual samples, that is, 10–33% and 8–16% in ESI− and ESI+ modes, respectively. Therefore, the large numbers of unique species between summer and winter indicate the large seasonal discrepancy in the sources and formation pathways of organic aerosols. Further details are discussed in Sections 3.3 and 3.4. Moreover, there were some common species identified in both ESI− and ESI+ modes during the study periods (33 in summer and 182 in winter, Fig. 4c), the numbers of which only accounted for 1–11% of total organic species in each mode for individual samples. Given that the detected organic molecular species were mostly distinct between the two ionization modes, combining the results together would provide complementary insights into the molecular characteristics of organic aerosols and gain an overall profile.

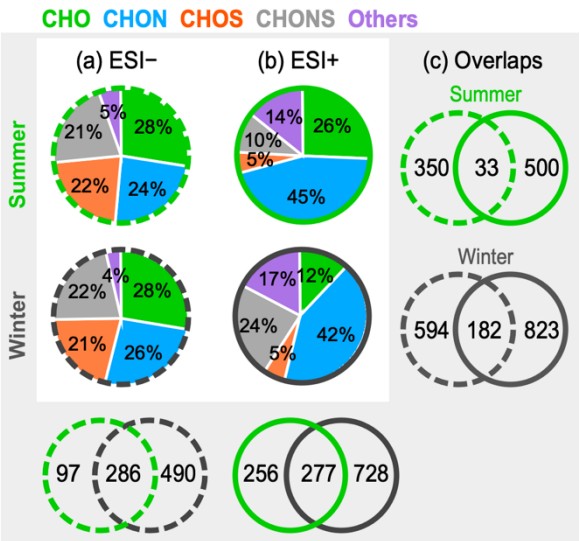

**Figure 4. Number fractions of each subgroup in total organic species detected respectively in (a) negative and (b) positive ESI modes during the summer and winter periods. (c) Numbers of common and unique organic species found in the two modes (ESI−: dashed circles, ESI+: solid circles) and in the two seasons (summer: green circles, winter: black circles), as presented in the surrounding gray area.**

### 3.2 Chemical characteristics of organic molecular species

Figure 5 presents the bulk chemical characteristics of organic molecular species identified in ESI− and ESI+ modes during the summer and winter periods. These parameters for individual organic subgroups are summarized in Table 2. In ESI− mode, although the molecular weights of organic species were mostly abundant at the range of approximately $m/z$ 168–424 for both seasons (as seen in the box and violin plots in Fig. 5a), the peak-area-weighted mean was higher in summer than those in winter ($m/z$ 244 vs. 220), with a shift of 24 Da. This is mainly attributed to the increased molecular weights of CHON, CHOS,

and CHONS species in summer (Table 2) that usually had lower volatilities (Li et al., 2016; Wang et al., 2024b). Similar variation patterns were also obtained in ESI+ mode, with a higher peak-area-weighted mean of molecular weight in summer than those in winter (*m/z* 247 vs. 208). Since air masses from the surrounding Qinling Mountains area in the south were more prevalent at the sampling site in summer season (Fig. 1b), biogenic emissions from forest vegetation (especially those with larger molecules such as monoterpenes and sesquiterpenes) and their oxidation products might be key factors resulting in the higher organic molecular weight, as evidenced by the presence of a number of biogenic tracer species such as MBTCA ($C_8H_{12}O_6$) and pinic acid ($C_9H_{14}O_4$). Other intensive processes such as thermal ablation of plant waxes in summer could also contribute to the higher molecular weight substances in the forest atmosphere (Deshmukh et al., 2019; Ehn et al., 2012).

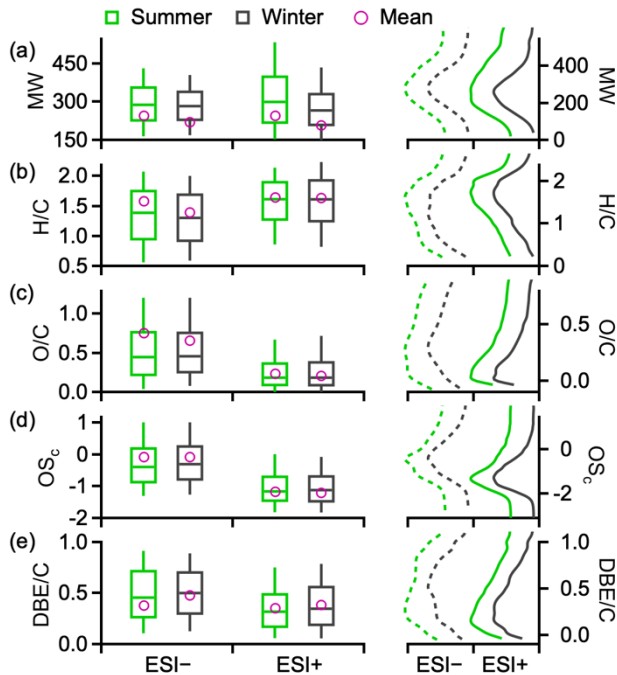

**Figure 5. Statistical analyses on the molecular characteristics of all organic species identified in summer and winter samples: (a) molecular weight, (b) H/C ratio, (c) O/C ratio, (d) the oxidation state of carbon atoms, and (e) ring and double-bond equivalent per carbon atom. The combined dataset of organic species identified in at least one sample in each season were used for analysis here. The key statistics and overall distribution profiles of these parameters were presented using the box (left) and violin plots (right), respectively. In the box plots, the bottom and top whiskers represent the 10th and 90th percentiles, respectively, the floating boxes represent the 25th–75th percentiles, and the short lines inside boxes represent the medians.**

**Table 2. Averaged species number, peak area intensity, and peak-area-weighted molecular weight (M.W.), elemental ratios (H/C and O/C), oxidation state of carbon atoms ($\overline{OS}_c$), ring and double-bond equivalent per carbon (DBE/C) for individual organic subgroups and total species assigned in negative and positive ESI modes over the studied summer and winter periods in 2021/2022.**

| Species | Summer | | | | | | | Winter | | | | | | |
| | number | peak area (10⁹) | M.W. | H/C | O/C | $\overline{OS}_c$ | DBE/C | number | peak area (10⁹) | M.W. | H/C | O/C | $\overline{OS}_c$ | DBE/C |
|---|---|---|---|---|---|---|---|---|---|---|---|---|---|---|
| *ESI−* | | | | | | | | | | | | | | |
| CHO | $419 \pm 51$ | $4.4 \pm 1.7$ | 206 | 1.45 | 0.62 | −0.20 | 0.41 | $617 \pm 49$ | $9.3 \pm 2.3$ | 211 | 1.28 | 0.50 | −0.29 | 0.48 |
| CHON | $380 \pm 117$ | $0.8 \pm 0.4$ | 250 | 1.37 | 0.49 | −0.38 | 0.51 | $592 \pm 107$ | $4.5 \pm 1.7$ | 192 | 1.03 | 0.54 | 0.04 | 0.71 |
| CHOS | $343 \pm 76$ | $3.8 \pm 2.4$ | 255 | 1.84 | 1.00 | 0.15 | 0.24 | $473 \pm 93$ | $5.2 \pm 2.2$ | 231 | 1.78 | 1.01 | 0.23 | 0.29 |

| | | | | | | | | | | | | | |
|---|---|---|---|---|---|---|---|---|---|---|---|---|---|
| CHONS | 340 ± 106 | 1.0 ± 0.4 | 302 | 1.63 | 0.84 | 0.06 | 0.44 | 503 ± 163 | 2.3 ± 1.3 | 265 | 1.67 | 0.78 | −0.10 | 0.41 |
| Others | 87 ± 32 | 0.2 ± 0.1 | 323 | 0.72 | n.a. | n.a. | 0.81 | 80 ± 16 | 0.1 ± 0.04 | 263 | 0.93 | n.a. | n.a. | 0.74 |
| Total | 1568 ± 370 | 10.1 ± 4.3 | 244 | 1.58 | 0.75 | −0.08 | 0.38 | 2266 ± 359 | 21.4 ± 5.8 | 220 | 1.39 | 0.65 | −0.09 | 0.47 |
| *ESI+* | | | | | | | | | | | | | | |
| CHO | 585 ± 88 | 9.4 ± 1.8 | 243 | 1.34 | 0.26 | −0.83 | 0.42 | 362 ± 45 | 8.8 ± 2.8 | 195 | 1.26 | 0.27 | −0.72 | 0.48 |
| CHON | 1040 ± 210 | 9.4 ± 2.2 | 234 | 1.87 | 0.27 | −1.33 | 0.31 | 1249 ± 145 | 17.8 ± 5.3 | 209 | 1.64 | 0.24 | −1.15 | 0.39 |
| CHOS | 118 ± 31 | 0.6 ± 0.2 | 406 | 1.68 | 0.23 | −1.23 | 0.22 | 161 ± 32 | 0.6 ± 0.3 | 321 | 1.52 | 0.18 | −1.17 | 0.32 |
| CHONS | 230 ± 58 | 0.7 ± 0.3 | 392 | 1.69 | 0.30 | −1.09 | 0.31 | 715 ± 130 | 4.8 ± 2.0 | 280 | 1.98 | 0.39 | −1.20 | 0.26 |
| Others | 325 ± 48 | 3.1 ± 0.9 | 177 | 1.58 | n.a. | n.a. | 0.42 | 525 ± 92 | 11.8 ± 6.0 | 175 | 1.71 | n.a. | n.a. | 0.36 |
| Total | 2298 ± 396 | 23.2 ± 4.1 | 247 | 1.64 | 0.23 | −1.18 | 0.35 | 3011 ± 417 | 43.8 ± 15.0 | 208 | 1.63 | 0.20 | −1.22 | 0.38 |

The carbon oxidation state $\overline{OS}_c$ and unsaturation degree (reflected by the DBE per carbon atom herein, DBE/C) derived from the H/C and O/C ratios of organic molecular species were further explored (Fig. 5b–e). The $\overline{OS}_c$ values of organic species in ESI− mode were mostly distributed at −0.9 to 0.3 range in the two seasons (Fig. 5d), which could result from various atmospheric oxidation processes such as photochemistry, heterogeneous reactions, and aqueous phase reactions (Bianchi et al., 2019; Kroll et al., 2011). In contrast, the $\overline{OS}_c$ values were much lower in ESI+ mode (−1.5 to −0.7 range), which were consistent with the more alkaline nature and/or reduced state of organic species detected in the positive mode (Lin et al., 2012a). The relatively higher H/C and O/C ratios eventually resulted in the slightly higher peak-area-weighted means of $\overline{OS}_c$ in summer than those in winter, that is, −0.08 vs. −0.09 in ESI− and −1.18 vs. −1.22 in ESI+ modes (Table 2). Moreover, the lower peak-area-weighted means and medians of DBE/C were obtained in summer for the two modes (Fig. 5e), indicating the more abundance of organic molecules with less unsaturated bonds or aromatic structures compared with those in winter. These results together revealed that organic species in this forest atmosphere were generally more oxidized and less unsaturated in summer than those in winter, which was possibly associated with the stronger photochemical processes as well as the less influence of anthropogenic pollutants, as evidenced by the higher $O_3$ while lower $SO_2$ and $NO_2$ concentrations in the summer period (Fig. 2c).

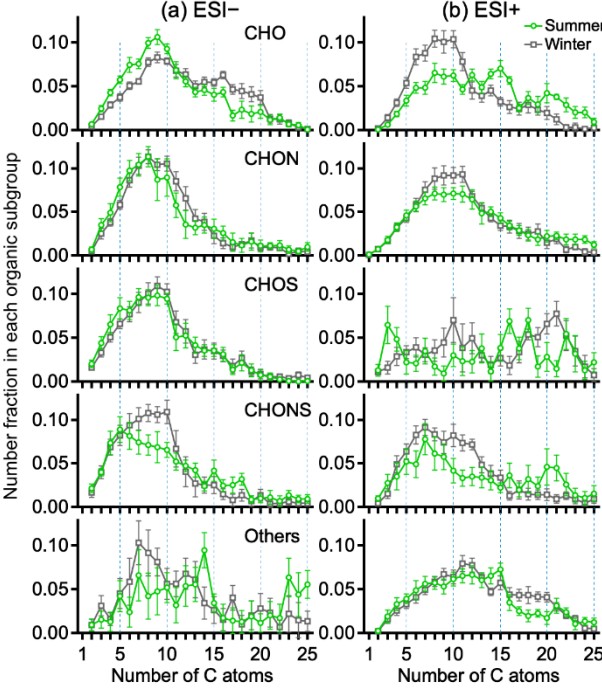

**Figure 6. Number fractions of organic subgroups sorted by the carbon atoms number in molecular formulas assigned in (a) negative and (b) positive ESI modes for summer and winter samples. Only organic species with $C_{\leq 25}$ are shown here due to their higher abundance. The error bars represent 1σ standard deviations for all the samples in each season.**

The observed changes in the molecular characteristics of organic species in the two seasons were as a result of the chemical variabilities of individual organic subgroups. Figure 6 presents the distributions of carbon atoms number in the molecular formulas of each subgroup during the study periods. The organic species with $C_2$–$C_{25}$ composed 82–99% in number (or 94–99% in peak area, Fig. S1 in the Supplement) of total species for individual subgroups in the two modes. Among them, the number and peak area fractions of organic species were particularly dominated by the $C_5$–$C_{11}$ compounds, which can be derived from both biogenic and anthropogenic sources. Moreover, the distributions of carbon atoms number for individual subgroups exhibited distinct seasonal variabilities. Specifically, the number fractions of CHO species with $C_{3-10}$ in ESI− mode were slightly higher in summer, whereas those with $C_{14-20}$ were apparently increased in winter. The abundance of different species in the two seasons could be closely associated with their origins and formation pathways (Wang et al., 2017a; Zhang et al., 2024). The distribution profiles of CHON and CHOS species in this mode were nearly comparable between summer and winter, indicating their similar sources and/or production pathways across the two seasons. In contrast, prominent increases were observed for the number fractions of CHO and CHON species in ESI+ mode and CHONS species in both modes at approximately $C_{6-11}$ range during the winter period, probably resulted mainly from the enhanced influence of anthropogenic pollution from the surrounding areas, compared with those of the summer period. Nevertheless, the degrees of influence could vary across individual subgroups, as indicated by the different variation patterns in their carbon atoms number distribution. The larger variations of CHONS species in both modes might result from the stronger influence of anthropogenic pollution compared with those of other species.

## 3.3 Contributions of anthropogenic and biogenic sources

A variety of organic tracer species of anthropogenic and biogenic origins reported by previous laboratory and field studies have been found in this forest atmosphere, the details of which are summarized in Table S1–S5 of the Supplement. They were primarily consisted of CHOS, CHONS, and CHO compounds in the ESI− mode, together with those of CHN and CHO compounds in the ESI+ mode. Figure 7 presents the temporal variations in the peak area intensities of these organic tracer species and their relative fractions over the study periods. Overall, distinct seasonal profiles were observed for the organic tracer species of different origins. The total peak area intensities of organic tracer species derived from biogenic precursors were almost comparable between the two seasons for both ESI− and ESI+ modes, whereas those derived from anthropogenic precursors were substantially increased in winter period (Fig. 7a, d). The ratios between the total peak area intensities of anthropogenic to biogenic tracer species in ESI− mode were on average 0.3 ± 0.1 in summer and 1.1 ± 0.2 in winter, corresponding to those of 0.4 ± 0.3 and 3.5 ± 1.7 in ESI+ mode. The large seasonal discrepancy in peak area intensity ratios of anthropogenic to biogenic tracer species suggests that organic aerosols over this forest atmosphere were more strongly contributed by anthropogenic sources in winter, contrasting with the more prevailing contribution of biogenic sources in summer.

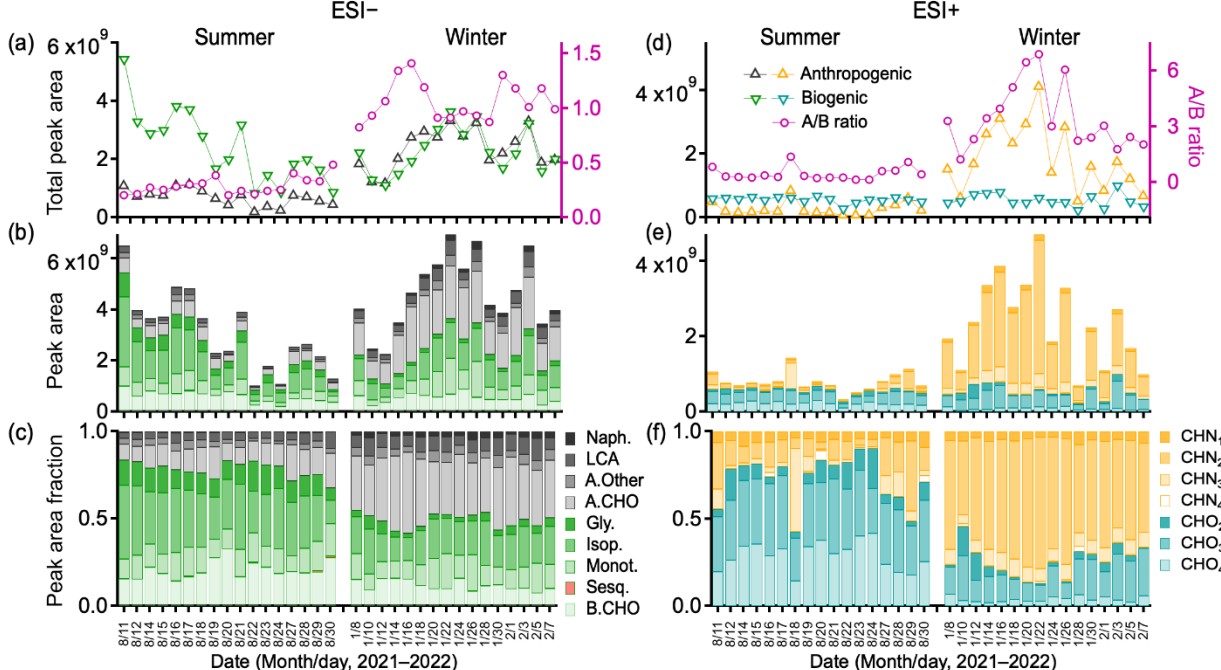

**Figure 7. (a, d)** Total peak area intensities of organic tracer species derived from anthropogenic and biogenic sources in negative and positive ESI modes, and the peak area ratios of anthropogenic to biogenic ones (A/B) over the study periods. **(b, c)** Peak area intensities of organic tracer species contributed by different types of precursors and **(e, f)** their relative fractions. In panels (b, c), the organic tracer species in ESI− mode consisted of various organosulfates, which were derived from anthropogenic (naphthalene, long-chain alkanes, and others) and biogenic (glyoxal, isoprene, monoterpenes, and sesquiterpene) precursors, and some other CHO compounds (Table S1–S3 in the Supplement). In panels (e, f), the organic tracer species in ESI+ mode consisted of CHN and CHO compounds derived from anthropogenic and biogenic sources, respectively (Table S4–S5).

The identified individual organic tracer species were mainly derived from biogenic precursors of glyoxal, isoprene, and monoterpenes along with anthropogenic precursors of naphthalene, long-chain alkanes, and others (Fig. 7b, e). In ESI− mode, the biogenic tracer species dominated the total peak area intensities in summer, accounting for 68–84% compared with those of 41–55% in winter (Fig. 7c). Among these, organosulfate and nitrooxy-organosulfate compounds (abbreviated as OSs hereafter), roughly defined as those with sufficient oxygen atoms to assign the −OSO$_3$H and −ONO$_2$ groups in the molecules (Brüggemann et al., 2020; Lin et al., 2012b), were important components in both seasons. Note that the designated OSs here would correspond to those at the upper limit, since no further fragmentation spectra were analyzed herein. The OSs derived from glyoxal and isoprene (mainly from biogenic sources) had higher peak area intensities and relative fractions in summer, which is consistent with the higher emissions and more intensive photochemical activities of these precursors than those in winter (Hu et al., 2017; Li et al., 2021; Vettikkat et al., 2023). On the contrary, the peak area intensities and fractions of monoterpenes derived OSs were on average slightly higher in winter than those in summer. Although the biogenic monoterpene emissions were likely higher in summer than in winter (Bai et al., 2017; Hakola et al., 2012), the highly abundance of anthropogenic pollutants (e.g., NO$_x$, SO$_2$, and sulfate; Fig. 2) would more facilitate the production of these OSs in winter. In addition, biogenic CHO tracer species were also important components of organic aerosols in ESI− mode, with comparable peak area intensities in the two seasons but higher fractions in summer (Fig. 7b, c). Similar trends were also observed for biogenic CHO tracer species in ESI+ mode, especially with the massive increase of C$_x$H$_y$O$_4$ fractions in summer (Fig. 7e, f). These CHO species were primarily associated with the oxidation products of isoprene and terpenes along with other biogenic species such as unsaturated fatty acids (Table S3 in the Supplement).

In contrast, the anthropogenic organic tracer species in both ESI− and ESI+ modes were relatively more abundant in winter, respectively accounting for 45–59% and 54–87% of total peak area intensities, compared with those of 16–32% and 10–57%

in summer. Among these, the OSs derived from anthropogenic precursors in ESI− mode (Table S2) had much higher peak area intensities and relative fractions in winter than those in summer (Fig. 7b, c). This was most likely attributed to the enhanced contribution from anthropogenic sources (e.g., combustion of biomass and fossil fuels) in the surrounding rural and urban areas during the winter heating season, together with the unfavorable atmospheric dispersion conditions (Cao and Cui, 2021; Zhao et al., 2015). A number of aromatic CHO species (such as benzoic acid $C_7H_6O_2$ and phthalic acid $C_8H_6O_4$, Table S3) were also found in ESI− mode and relatively more abundant in winter, which might be contributed by primary combustion emissions and/or secondary products of anthropogenic aromatic precursors (Aggarwal and Kawamura, 2008; Vodička et al., 2023). Moreover, a variety of oxygen-free nitrogen-containing CHN species were identified as the major anthropogenic tracer species in ESI+ mode (Table S5). Their peak area intensities were dominated by the $C_xH_yN_2$ compounds such as $C_5H_8N_2$, $C_6H_{10}N_2$, $C_7H_{12}N_2$, $C_8H_8N_2$, and $C_9H_{10}N_2$ (Fig. 7f). These $C_4H_6N_2$-$(CH_2)_n$ and $C_6H_4N_2$-$(CH_2)_n$ homologs have been reported as heterocyclic alkaloids derived from coal combustion and biomass burning (Li et al., 2024c; Song et al., 2022; Wang et al., 2017b).

The anthropogenic and biogenic organic tracer species identified in the ESI− and ESI+ modes were further analyzed in the van Krevelen diagram based on their elemental ratios of O/C and H/C, as shown in Fig. 8a. Similar variation patterns were generally observed between the two modes. Specifically, except for those derived from glyoxal, isoprene, and some anthropogenic precursors with O/C > 0.5, other tracer species were primarily distributed in the regime with O/C ≤ 0.5. In addition, the anthropogenic tracer species widely dispersed at the regime with H/C from approximately 0.5 to 2.5, whereas the biogenic ones were mainly situated at the H/C ≥ 1.5 regime. This result is generally consistent with previous findings that the regime with H/C ≥ 1.5 and O/C ≤ 0.5 mainly consists of aliphatic species of anthropogenic and biogenic origins, while the regime with H/C ≤ 1 and O/C ≤ 0.5 are dominated by aromatic species of anthropogenic origins (Kourtchev et al., 2016a; Lin et al., 2022; Wang et al., 2017a). These two regimes are respectively represented as the pink and blue shaded areas in Fig. 8.

The organic molecular species assigned on individual days during the study periods were then compared in the aliphatic and aromatics regimes of the van Krevelen diagram. Figure 8b presents the example results obtained in the ESI− mode on two different days (that is, August 30 and January 26) for illustration. A larger number of organic species were present in the aliphatic regime than those in the aromatic regime (310 vs. 175) on August 30 in summer, whereas enhanced aromatic species (449 vs. those of 629 in the aliphatic regime) were observed on January 26 in winter. Also, more organic species were found in both aliphatic and aromatic regimes on the winter day compared with the summer day, suggesting the enhanced influence of anthropogenic pollution in winter over this forest atmosphere. A statistical analysis was further performed for all the summer and winter samples detected in both ESI− and ESI+ modes across the entire study periods, as shown in Figure 8c, from which similar seasonal variation tendencies were also obtained as those of the above two example days. In ESI− mode, the number fraction of organic species in the aromatic regime was on average 21 ± 2% on individual days in winter, which was higher than those of 17 ± 2% in summer. The peak area fractions of these aromatic species were even 2.7 times higher in winter than those in summer (25 ± 4% vs. 9 ± 2%). On the contrary, both the number and peak area fractions of aliphatic species were higher in summer than those in winter. Likewise, in ESI+ mode, the number and peak area fractions of aromatic species were slightly higher in winter than in summer, while those of aliphatic species were higher in summer. These results further confirmed the dominant roles of biogenic sources in summer but those of anthropogenic sources in winter in the production of organic aerosols over this forest atmosphere.

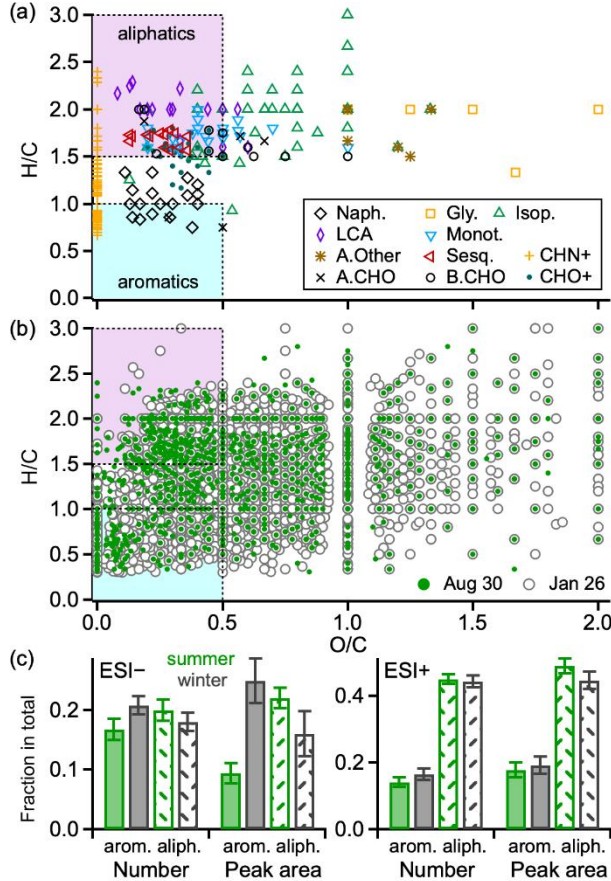

**Figure 8.** van Krevelen diagrams plotting of (a) organic tracer species derived from anthropogenic and biogenic precursors in both modes (as listed in Table S1–S5), and (b) organic species identified on a summer day (August 30) and a winter day (January 26) in ESI− mode. (c) Averaged number and peak area fractions of organic species in aromatic and aliphatic molecular structures for summer and winter samples in ESI− and ESI+ modes. In panels (a, b), the shaded pink (H/C ≥ 1.5, O/C ≤ 0.5) and cyan (H/C ≤ 1.0, O/C ≤ 0.5) areas represent aliphatic and aromatic regimes, respectively. The O/C was replaced by (O−3S)/C or (O−3S−2N)/C in the case of organosulfates or nitrooxy-organosulfates. The error bars in panel (c) represent 1σ standard deviations for all the samples in each season.

## 3.4 Synergistic effects of meteorology and anthropogenic pollutants

The production and molecular composition of organic aerosols can be affected by multiple environmental factors in this forest atmosphere. Among those, ambient RH turned out to be one of the key governing factors. Figure 9 presents the species number and fraction of individual organic subgroups in ESI− mode under varied RH conditions during the study periods. With the mean RH values increasing from $62 \pm 6\%$ to $75 \pm 4\%$ in summer, some evident increases (1.1–1.2 times, Fig. 9a) were observed in the numbers of CHO, CHON, CHOS, and CHONS species, possibly due to the enhanced heterogeneous and/or aqueous phase reactions of aerosol particles in semisolid or liquid physical state under higher RH conditions (Ervens et al., 2011; Reid et al., 2018; Wang et al., 2021d). Meanwhile, the number fractions of CHO species were decreased while those of other subgroups were increased (Fig. 9b). It was likely that more anthropogenic gas pollutants ($SO_2$, $NO_x$, and etc.) could absorb into aerosol liquid phase at the higher RH conditions, reacting with preexisting CHO species such as hydroperoxides, carbonyl/hydroxyl compounds, and unsaturated fatty acids to produce more nitrogen- and/or sulfur-containing species (Wang et al., 2021e; Xu et al., 2021c; Zhu et al., 2019). Nevertheless, all the species numbers were reversely decreased at the extremely high RH levels (mean $93 \pm 2\%$) in summer, during which heavy rainfalls occurred (Fig. 2d). Aerosol particles can be easily washed away under such heavy precipitation conditions. By comparison, the observed RH effect in winter was generally

similar to those in summer, except without heavy precipitation. The numbers and relative fractions of organosulfur species

CHOS and CHONS increased constantly for RH increased from $43 \pm 8\%$ to $88 \pm 5\%$ (Fig. 9c, d). The decreased number and fraction of CHON species at the highest RH level in winter could partly result from their hydrolysis reactions (Hu et al., 2011; Su et al., 2021). In contrast, the similar but weaker dependence on RH was found for organic species in the ESI+ mode (Fig. S2 in the Supplement), probably associated in part with their distinct physicochemical properties compared with those in the ESI− mode. This result also indicates that the influence of RH could be stronger for acidic organic substances, since these

were more readily deprotonated and sensitive to be detected in negative mode than the basic ones (Laskin et al., 2018).

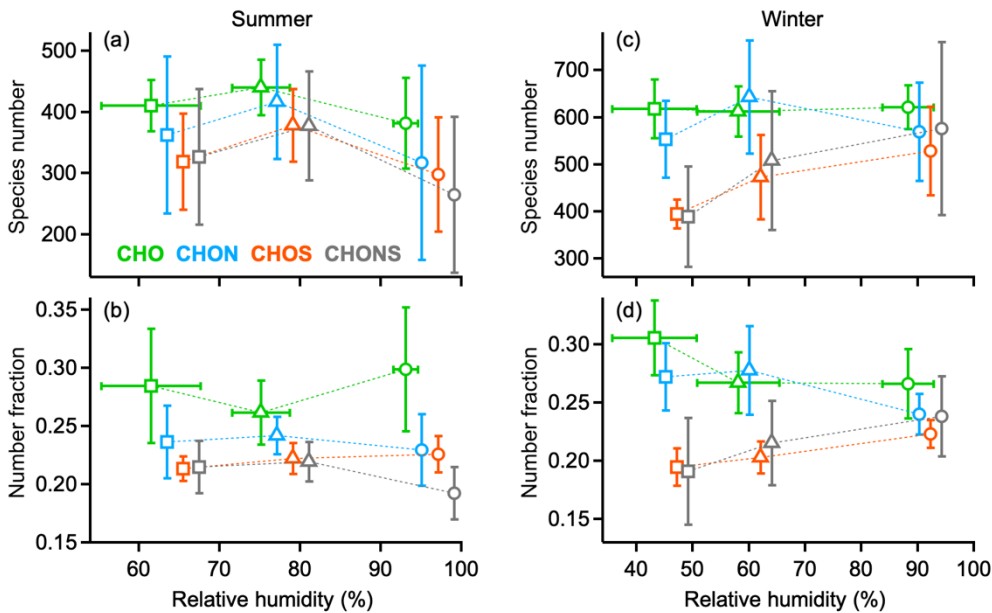

**Figure 9. Species number and the relative fractions of organic molecular subgroups in negative ESI mode under varied**
**relative humidity conditions in (a, b) summer and (c, d) winter periods. The symbols of each type shared the same RH values (that is, $62 \pm 6\%$, $75 \pm 4\%$, $93 \pm 2\%$ in summer and $43 \pm 8\%$, $58 \pm 7\%$, $88 \pm 5\%$ in winter) but were offset horizontally for a clear vision. The horizontal error bars in the CHO species represent 1σ standard deviations of RH values. The vertical error bars represent 1σ standard deviations for all the samples in each RH condition.**

On the contrary, the effect of ambient temperature on organic molecular composition was more complex over the study periods. The species numbers of all organic subgroups in ESI− mode were almost comparable at the mean temperatures of $21.5 \pm 0.8$ and $24.5 \pm 0.8$ °C in summer, while they were largely increased at the highest temperature level of $27.6 \pm 1.0$ °C (Fig. S3 in the Supplement). In contrast, all the species numbers were decreased at the highest temperature in winter ($3.8 \pm 1.0$ °C). Their number fractions also had no obvious variation trends in the two seasons. The species number and fraction of organic subgroups

in ESI+ mode varied insignificantly with temperature as well (Fig. S4 in the Supplement). Higher temperatures usually would promote biogenic volatile organic emissions from forest vegetation to form more secondary products, especially during summertime (Chang et al., 2014; Kourtchev et al., 2016b). The lack of a strong temperature dependence herein suggests that other influencing factors might be more dominant or the temperature variability in each season was too small to result in significant changes on the reaction chemistry of organic aerosols during the study periods.

A variety of anthropogenic pollutants also played crucial roles in affecting organic aerosol composition in this forest atmosphere. The winter period was primarily characterized by the higher loadings of anthropogenic pollutants $NO_2$, $SO_2$, sulfate, and nitrate compared with those in summer (Fig. 10a, b). The nitrogen and sulfur oxidation ratios (NOR and SOR) were also higher in winter, whereas the atmospheric oxidation capacity was stronger in summer (as indicated by the higher

odd oxygen levels) (Fig. 10c). Some typical organic tracer species were identified under these different atmospheric conditions in each season. For instance, several organosulfates (e.g., $C_3H_6O_5S$, $C_3H_6O_6S$, and $C_4H_8O_6S$) were found to have the highest peak area intensities in winter samples, which were most likely derived from isoprene oxidation under high $NO_x$ conditions (Schindelka et al., 2013; Shalamzari et al., 2013). In contrast, some isoprene oxidation products under low $NO_x$ conditions, such as $C_5H_{12}O_7S$ and $C_5H_{10}O_7S$ (Surratt et al., 2008, 2010), were commonly found in summer samples. Moreover, a monoterpenes-derived organosulfate of $C_{10}H_{17}NO_7S$ was greatly abundant in both summer and winter samples. It can be generated either through the photooxidation of monoterpenes in the presence of $NO_x$ and $SO_2$ or their nocturnal chemistry initiated by nitrate radicals (Iinuma et al., 2007; Surratt et al., 2008).

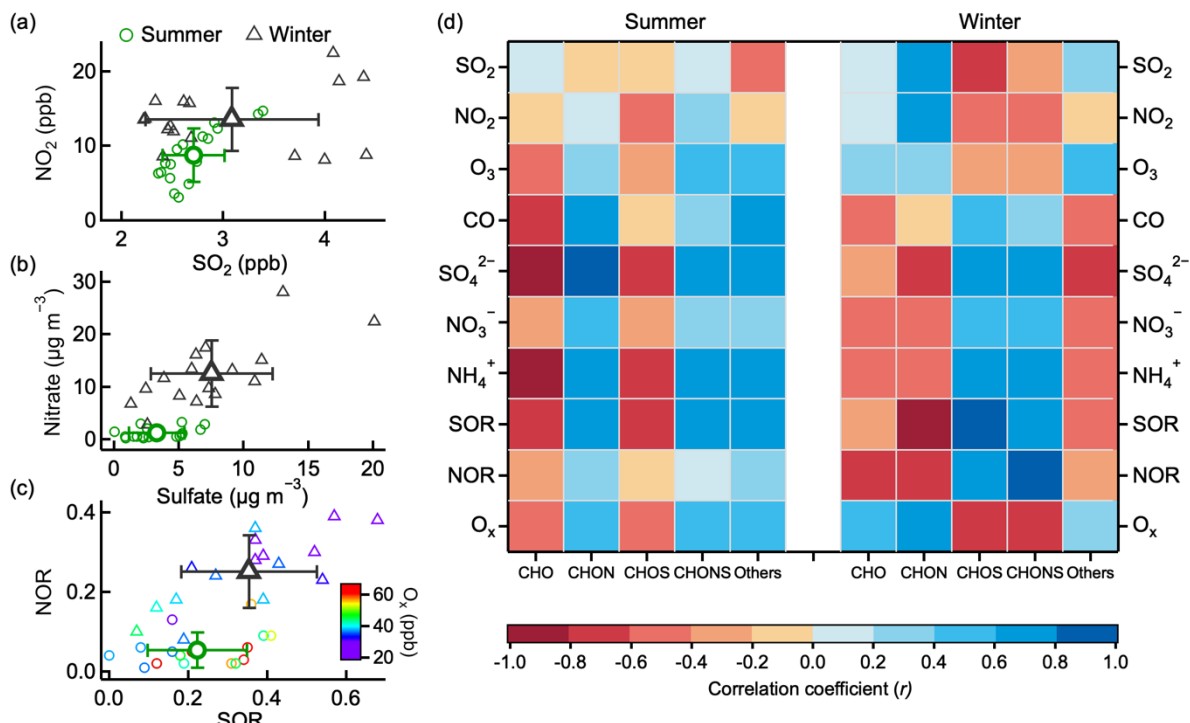

**Figure 10. Typical atmospheric conditions of (a) NO₂ and SO₂, (b) nitrate and sulfate, and (c) nitrogen and sulfur oxidation ratios and oxidant levels in the studied summer and winter periods. (d) Correlations of gas and particulate pollutants with the number fractions of organic subgroups in negative ESI mode, which were all statistically highly significant at *p* < 0.001 level. In panels (a–c), the larger symbols with error bars represent the mean values and 1σ standard deviations of these parameters in each season.**

The effects of anthropogenic pollution on the molecular composition of organic aerosols were further explored to understand their potential interactions. Figure 10d presents the correlations of gas and particulate pollutants, SOR, NOR, and $O_x$ with the number fractions of individual organic subgroups in ESI− mode over the study periods. Distinct correlation patterns were obtained between summer and winter, partly attributed to the different production pathways of organic aerosols under the synergistic effects of anthropogenic pollutants and meteorology in the two seasons. In the summer period, the number fractions of CHO and CHOS species negatively correlated with all these parameters (correlation coefficients *r*: −0.07 to −0.82), except for a weak positive correlation of CHO species with $SO_2$ (*r*: 0.17). This phenomenon could partly result from the chemical transformation of these species into other chemically diverse substances (e.g., CHON and CHONS) through acid-catalyzed heterogeneous reactions and/or aqueous phase chemistry in the presence of anthropogenic air pollutants (Darer et al., 2011; Fan et al., 2022; Kwong et al., 2018; Lam et al., 2019). Another proposed explanation is that the productions of CHO and CHOS species were to some extent governed by the consumption of anthropogenic pollutants at the low concentration levels

in summer (Fig. 2c), as those reported in some other ambient atmospheres (Lin et al., 2022; Riva et al., 2019; Wang et al., 2020). On the contrary, the CHON, CHONS, and other oxygen-free species (mainly CHNS herein) positively correlated with most of these parameters ($r$: 0.16–0.81; except for $SO_2$ and $NO_2$), suggesting their large anthropogenic contributions. Possible reaction pathways include such as biogenic isoprene and monoterpene precursors being oxidized by nitrate radicals to form condensable organic nitrates and nitrooxy-organosulfates (Hamilton et al., 2021; Ng et al., 2017). These results revealed that anthropogenic pollutants also played important roles in affecting the chemical composition of organic aerosols in summer period, despite their relatively low concentrations.

In winter period, however, the number fractions of CHO and CHON species correlated positively with $SO_2$, $NO_2$, $O_3$, and $O_x$ but negatively with other parameters. These species might be partly associated with primary emissions (Song et al., 2018; Zhong et al., 2023), such as those of biomass burning and fossil fuel combustion from the surrounding rural and urban areas. These correlation patterns were just opposite to those of organosulfur species (CHOS and CHONS). The strong positive correlations of organosulfur species with SOR and NOR ($r$: 0.60–0.86) suggest that secondary formation was important processes for their production. Indeed, several secondary reaction mechanisms have been previously proposed for organosulfur species formation, including acid-catalyzed ring-opening of epoxides, $SO_2$ reacting with unsaturated hydrocarbons, direct sulfate esterification, sulfate radical reactions in the aqueous phase, and so on (Brüggemann et al., 2020; Fan et al., 2022). Nevertheless, the atmospheric oxidants appeared to be a limiting factor for those reactions under the studied winter conditions, as indicated by the high negative correlations of $O_x$ with the CHOS and CHONS species ($r$: –0.64 and –0.66). This is consistent with previous studies that oxidant levels were one of the most important governing factors for the production of organosulfur species by limiting the oxidation of organic precursors (Bryant et al., 2021; Wang et al., 2021f). Taken these observations together, the synergistic effects of meteorological factors and anthropogenic pollutants have played critical roles in affecting the production pathways and chemical transformation of organic aerosols, thereby altering their molecular composition and other related properties under different seasonal conditions in this forest atmosphere.

## 4 Summary and Implications

The molecular composition of organic aerosols in atmospheric $PM_{2.5}$ was characterized using UHPLC–HRMS at a forest site in the Qinling Mountains region of central China during the two contrasting seasons (summer and winter) of 2021/2022. Organic molecular species assigned in the negative and positive ESI modes had much higher numbers and peak area intensities in winter than those in summer, which were consistent with the mass concentration trends of total organic matter, suggesting that organic aerosols were more abundant and chemically diverse in this forest atmosphere during the wintertime. The higher peak-area-weighted mean values of molecular weight and oxidation state but the lower unsaturation degree of organic species were observed in summer, possibly resulted from the large biogenic emissions from forest vegetation and intense photochemical processes. The relative abundances of a variety of identified organic tracer species revealed that organic aerosols were substantially affected by anthropogenic sources in winter, whereas biogenic sources were more prevalent in summer. The increased organic species with aliphatic and aromatic molecular structures in the winter period were closely associated with the oxidation products of anthropogenic long-chain alkanes and aromatic hydrocarbons transported from the surrounding rural and urban areas. The production and transformation of organic aerosols in this region were largely influenced by the synergistic effects of meteorology and anthropogenic pollutants, which ultimately altered their molecular composition and other characteristics. The current work focuses primarily on the non-target characterization of organic species in aerosol particles over the Qinling Mountains region based on the HRMS analysis. To further identify organic tracers and other specific substances with the aid of fragmentation spectra in existing MS databases would achieve even higher reliability. Future studies on the quantification and source apportionment of organic molecular composition will be also valuable to elucidate the complex anthropogenic–biogenic interactions and to incorporate the results into atmospheric chemistry model for improved

prediction of organic aerosol burden and impacts in similar regions worldwide.

**Data Availability.** The datasets used in this study are available upon request to the corresponding author.

**Supplement.** The supplement related to this article is available online at: https://doi.org.

**Author contributions.** YH and JC designed and conceptualized the study. XZ with support from JL, YC, CZ performed aerosol sampling and field measurements. XZ, LL, YL, RW, and SX contributed to the aerosol chemical analysis and data processing. XZ and YH wrote the original draft. All authors revised and approved the manuscript before submission.

**Competing interests.** The contact author has declared that none of the authors has any competing interests.

**Acknowledgements.** The authors gratefully thank the support and assistance from the staff of the National Observation and Research Station of Regional Ecological Environment Change and Comprehensive Management in the Guanzhong Plain.

**Financial support.** This work has been supported by the National Natural Science Foundation of China (42177094 and 42007207), the Program of Chinese Academy of Sciences (292020000018), and the Sino-Swiss cooperation project on Clean Air China (7F-09802.02.01).

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
