# Peer review of "Influence of anthropogenic pollution on the molecular composition of organic aerosols over a forest site in the Qinling Mountains region of central China"

_EGUsphere, 2025_

## Author Comment (AC1)

**Response to the comments of Referee #1 on "Influence of anthropogenic pollution on the molecular composition of organic aerosols over a forest site in the Qinling Mountains region of central China"**

by Xin Zhang and Yuemei Han

We greatly appreciate the valuable comments and suggestions from Referee #1 on our manuscript (ID: egusphere-2025-519). We have carefully revised the manuscript by taking into account of all the concerns raised by the referee. Please find our detailed point-to-point responses to these comments below. A copy of the manuscript and supplement with all the changes and other minor corrections tracked is also attached for reference.

(*The blue bold, green, and black fonts represent the Referee's comments, the related text in the manuscript, and the authors' responses, respectively.*)

*In this study, Zhang et al. report an extensive molecular characterisation of organic aerosol in particulate matter (PM$_{2.5}$) from a forest site in the Qinling mountains, China, and investigated the influence of anthropogenic pollution on the aerosol composition. For this they sampled 33 filter samples in summer and wintertime, used liquid extraction and measured the extracts with an ultrahigh-performance liquid chromatography coupled to high-resolution Orbitrap mass spectrometry. With a non-target analysis known and unknown compounds were detected and identified. The combination of molecular fingerprints, air quality measurements, meteorological data as well as back-trajectories enables detailed interpretation of origin and transformation pathways of the sampled aerosol. Tracer species confirm the interpretation of more influence from biogenic precursors in summer and a more diverse and largely anthropogenic influenced composition in winter.*
*The authors have combined their comprehensive results and have produced a detailed characterization of the molecular composition of organic aerosol. The manuscript has a*

*good structure, meaningful illustrations and is written clear and precise. The work is worth to be published in "Atmospheric Chemistry and Physics" with some minor comments.*

Response: We gratefully thank Referee #1 for the insightful and positive evaluation on our manuscript. We have seriously addressed all the concerns and revised the manuscript accordingly. Our detailed responses to these comments are as follows.

*Comments*

*1. L84: Delete "city" in "[…] 50 km southwest of the megacity city Xi'an, as shown […]".*

Related sentence: "This site is situated in the northern foothill of the Qinling Mountains and approximately 50 km southwest of the megacity **city** Xi'an, as shown in Fig. 1."

Response: Yes, it has been deleted in the revised manuscript, as follows:

"This site is situated in the northern foothill of the Qinling Mountains and approximately 50 km southwest of the **megacity Xi'an**, as shown in Fig. 1." (Lines 83–84)

*2. L102: Have the authors considered that ultrasonication can influence the chemical composition due to free radical production? (Miljevic et al., 2014)*

Related sentence: "A quarter of each sample in 12.56 cm$^2$ area was ultrasonically extracted using 9 g acetonitrile and water mixture in 9:1 volume for 30 min (3 g for 10 min, repeated three times)."

Response: Yes, we definitely agree with the referee that the ultrasonication of filter samples can degrade the chemical composition due to OH radical production under certain conditions. The chemical effects of ultrasonic irradiation mainly resulted from the high local temperature and pressures within the collapsing cavitation bubbles by the energy of ultrasonic waves (Miljevic et al., 2014). However, given that we performed the ultrasonic extraction of filter samples in a water–ice bath and the procedure lasted 10 min for each time in this study, the chemical effects should be limited under the low temperatures.

Moreover, the effects of ultrasonication on aerosol chemical composition were turned out to depend largely on the extraction solvents. As demonstrated by Hettiyadura et al. (2015), for

higher percentage of organic solvents (that is, acetonitrile and ultra-pure water in 95 : 5 by volume in their study), there were no degradation effects and only minor chemical differences between the ultrasonication and rotary shaking procedures for sample treatment. In fact, the ultrasonication was determined to be a better method due to its higher precision compared with rotary shaking (Hettiyadura et al., 2015). Similarly, we used the solvent of acetonitrile and pure water mixture in 9 : 1 volume in our current study, thereby the chemical effects should be at least not significant.

Taken the above two aspects together, we believe that the chemical composition should be insignificantly affected by the ultrasonication procedure in this study. We have highlighted this point in the revised manuscript, as follows:

"The extraction system was placed in a water–ice bath to **eliminate potential evaporation or chemical reactions** of aerosol components." (Lines 103–104).

Two references cited above are listed below:

Miljevic, B., Hedayat, F., Stevanovic, S., Fairfull-Smith, K. E., Bottle, S. E., and Ristovski, Z. D.: To sonicate or not to sonicate PM filters: Reactive oxygen species generation upon ultrasonic irradiation, Aerosol Sci. Technol., 48, 1276–1284, 2014.

Hettiyadura, A. P. S., Stone, E. A., Kundu, S., Baker, Z., Geddes, E., Richards, K., and Humphry, T.: Determination of atmospheric organosulfates using HILIC chromatography with MS detection, Atmos. Meas. Tech., 8, 2347–2358, 2015.

*3. L142: Calculating HYSPLIT trajectories at 34.06° N, 108.34° E, 500 m height above ground level result in a height of 1885 m above sea level, since the height of the cell grid is already 1385 m. Can the authors comment on why they used 500 m height above ground level?*

Related sentence: "In addition, the backward trajectories of air masses **(500 m height above ground level** and 72 h duration) arrived at the sampling site were calculated hourly across the study periods using the Hybrid Single Particle Lagrangian Integrated Trajectory model (v5.2.1) (Stein et al., 2015)."

Response: The altitude of our sampling site is around 530 m above the sea level, as described in the manuscript, "Atmospheric $PM_{2.5}$ was sampled at a forest site (34.06° N, 108.34° E,

around 530 m above sea level) in the Qinling Mountains region of central China during summer and winter seasons of 2021/2022." This is somewhat different from the height of the cell grid (that is, 1385 m) suggested by the referee, which was possibly obtained from a different method. We used the 500 m height above ground level for calculating the back trajectories of air masses, mainly considering the requirement of HYSPLIT model and the height of planetary boundary layer in our studied area. On the one hand, it is not recommended to start the model at the height too close to the ground, because the trajectory can easily hit the ground and lose accuracy, as demonstrated by the HYSPLIT Cheat Sheet (https://www.ready.noaa.gov/documents/ppts/Cheat_Sheet_2020.pdf). On the other hand, the 500 m height is mostly situated within the planetary boundary layer and close to the height of mixed layer in our studied area, according to the results reported in previous literature (e.g., Xin et al., 2024; Wang et al., 2023). This height thus could reflect the average concentrations of air pollutants in the well-mixed boundary layer, which is suitable for investigating the transport of near-surface pollution from the surrounding urban and rural areas to the studied location. Therefore, we used the 500 m above ground level to calculate the back trajectories of air masses in our current study.

Two references cited above include:

Xin, J., Peng, K., Zhu, X., Pan, X., Wang, Q., Cao, J., Wang, Z., Cao, X., Ren, X., Yang, S., Wei, Y., Zhao, D., and Ma, Y.: AI model to improve the mountain boundary layer height of ERA5, Atmos. Res., 304, 2024.

Wang, Y., Xu, T., Shi, G., Yang, F., Tang, X., Zhao, X., Wan, C., and Liu, S.: Climatology of the planetary boundary layer height over China and its characteristics during periods of extremely temperature, Atmos. Res., 294, 2023.

*4. L205: Can the compounds also have different transmission efficiencies in the mass spectrometer? And is therefore not just an effect of different ionization efficiencies?*

Related sentence: "This result can be attributed to either the large abundance of these species in this forest atmosphere or their high **ionization efficiency** under ESI− mode, despite that quantitative analyses were not available herein due to the technical limitation."

Response: The referee raised a good point here. We agree that organic compounds with different chemical properties (e.g., molecular size, structure, and polarity) would have varied transmission efficiencies in the mass spectrometer. Therefore, both the ionization and transmission efficiencies could affect the observed peak area intensity of organic species. We have added this information in the revised manuscript, as follows:

"This result can be attributed to either the large abundance of these species in this forest atmosphere or their high **ionization and transmission efficiencies** under ESI− mode, despite that quantitative analyses of the highly complex composition of organic aerosols remain challenging due to the lack of authentic standards (Evans et al., 2024; Ma et al., 2022; Noziere et al., 2015)." (Lines 204–207)

*5. L206: Can the authors clarify what they mean with "technical limitations"?*
*The lack of authentic standards (and the fact that many compounds in ambient PM are not even precisely characterized) does not allow a quantitative approach on such a highly complex composition, independent of the analytical devices used. For example, Ma et al. (2022) and Evans et al. (2024) showed that semi-quantification is possible but with great uncertainties.*

Related sentence: "This result can be attributed to either the large abundance of these species in this forest atmosphere or their high ionization efficiency under ESI− mode, despite that quantitative analyses were not available herein due to the **technical limitation**."

Response: As suggested by the referee, we have clarified the meaning of "technical limitations" in the revised manuscript, as follows:

"This result can be attributed to either the large abundance of these species in this forest atmosphere or their high ionization and transmission efficiencies under ESI− mode, despite that quantitative analyses **of the highly complex composition of organic aerosols remain challenging due to the lack of authentic standards (Evans et al., 2024; Ma et al., 2022; Noziere et al., 2015)**." (Lines 204–207)

The following two references have been also added in the revised manuscript:

"**Evans, R. L., Bryant, D. J., Voliotis, A., Hu, D., Wu, H., Syafira, S. A., Oghama, O. E.,**

McFiggans, G., Hamilton, J. F., and Rickard, A. R.: A Semi-Quantitative Approach to Nontarget Compositional Analysis of Complex Samples, Anal. Chem., 96, 18349–18358, https://doi.org/10.1021/acs.analchem.4c00819, 2024." (Lines 582–584)

"Ma, J., Ungeheuer, F., Zheng, F., Du, W., Wang, Y., Cai, J., Zhou, Y., Yan, C., Liu, Y., Kulmala, M., Daellenbach, K. R., and Vogel, A. L.: Nontarget Screening Exhibits a Seasonal Cycle of PM2.5 Organic Aerosol Composition in Beijing, Environ. Sci. Technol., 56, 7017–7028, https://doi.org/10.1021/acs.est.1c06905, 2022." (Lines 693–695)

*6. Figure 3: The legend is not optimally visible in panel b. Since the legends applies to all panels, maybe a central positioning above would be easier to see (compare with the legend of Fig. 4).*

Response: Yes, we agree. The legend of Figure 3 has been placed at the top central in the revised manuscript. Now it is clearly visible for all panels, as follows:

[Figure]

**Figure 3.**

*7.1 Figure 5: In the caption the authors explain " […] dashed lines inside boxes […]", but the lines are solid.*

Related sentence: "In the box plots, the bottom and top whiskers represent the 10th and 90th percentiles, respectively, the floating boxes represent the 25th–75th percentiles, and the **dashed lines** inside boxes represent the medians."

Response: Here we meant the "short lines" indeed. Therefore, we have made the following correction in the revised manuscript:

"In the box plots, the bottom and top whiskers represent the 10th and 90th percentiles, respectively, the floating boxes represent the 25th–75th percentiles, and the **short lines** inside boxes represent the medians." (Lines 267–269)

*7.2 Can the authors explain why they show box plots and violin plots? Since the data is not bimodal distributed, the reader can hardly get any additional information from both plots.*

Response: The purpose of showing both box plots and violin plots in Figure 5 is because of the following reasons. The box plots provided the key statistic summaries of data, including the values of mean, median, maximum, minimum, and percentiles. In contrast, the violin plots presented a more intuitive depiction of the data distribution from probability density across different values. Combining these two together can provide more comprehensive illustrations of the key statistics and overall distributional profiles of the data. Therefore, we have highlighted this point in the revised manuscript, as follows:

"The **key statistics and overall distribution profiles** of these parameters were presented using the box (left) and violin plots (right), respectively." (Lines 266–267)

*8. L311: The comparison of tracer species was only made based on the sum formula? Have the authors considered fragmentation spectra to identify compounds with databases to get a higher level of confidence (e.g. aerosolomics (Thoma et al. (2022)) or the mzCloud database)?*

Related sentence: "A variety of organic tracer species of anthropogenic and biogenic origins reported by previous laboratory and field studies have been found in this forest atmosphere, the details of which are summarized in Table S1–S5 of the Supplement."

Response: Since we performed the UHPLC–HRMS analysis in the full scan mode, there were no fragmentation spectra available in this current study. Also, for most of these hundreds of organic tracer species, we could not find their fragmentation spectra in the original literature. Therefore, a parallel comparison of these species with other studies was primarily based on

the sum formula. The scan mode information of our study has been added in the revised manuscript as: "The mass range was set at $m/z$ 50–750 **in full MS scan mode**, with the mass resolution of approximately 140,000 at $m/z$ 200." (Lines 114–115).

Nevertheless, we definitely agree with the referee that identifying compounds with the support of available fragmentation spectra from existing databases can get a higher level of confidence. To analyze organic tracer species and other specific compounds using HRMS, it would be more accurate to perform a full MS scan followed by MS/MS fragmentation scans of all ions at a wide isolation range. This should be an important topic in our future study. As a result, we have proposed this research topic in the Summary and Implications section of the revised manuscript, as follows:

"**The current work focuses primarily on the non-target characterization of organic species in aerosol particles over the Qinling Mountains region based on the HRMS analysis. To further identify organic tracers and other specific substances with the aid of fragmentation spectra in existing MS databases would achieve even higher reliability. Future studies** on the quantification and source apportionment of organic molecular composition **will be also valuable** to elucidate the complex anthropogenic–biogenic interactions and to incorporate the results into atmospheric chemistry model for improved prediction of organic aerosol burden and impacts in similar regions worldwide." (Lines 507–512)

*9. L336: Have fragmentation experiments carried out to validate the assignment of the compound classes? The functional groups of organosulfates (m/z 96.9601 (HSO₄⁻) and m/z 79.9573 (SO₃⁻)) as well as nitrate groups (m/z 61.9883 (NO₃⁻)) are very strongly represented in the fragmentation spectra and a clear indicator for organosulfates and nitrooxy organosulfates.*

Related sentence: "Among these, organosulfate and nitrooxy-organosulfate compounds (abbreviated as OSs hereafter), defined as those with sufficient oxygen atoms to assign the $-OSO_3H$ and $-ONO_2$ groups in the molecules (Brüggemann et al., 2020; Lin et al., 2012b), were important components in both seasons."

Response: As described in the response to comment #8, the HRMS analysis was performed only in the full MS scan mode, and thereby no fragmentation spectra are available for this study. In fact, it was a commonly used method in previous literature to identify organosulfate and nitrooxy-organosulfate compounds based on the assignment of $-OSO_3H$ and $-ONO_2$ groups from HRMS analysis, although we did not carry out fragmentation experiments to validate the assignment of these classes herein. Nevertheless, we agree that this method would result in an upper limit of the predicted numbers. Therefore, we have revised/added the following statement regarding this uncertainty in the revised manuscript:

"Among these, organosulfate and nitrooxy-organosulfate compounds (abbreviated as OSs hereafter), **roughly** defined as those with sufficient oxygen atoms to assign the $-OSO_3H$ and $-ONO_2$ groups in the molecules (Brüggemann et al., 2020; Lin et al., 2012b), were important components in both seasons. **Note that the designated OSs here would correspond to those at the upper limit, since no further fragmentation spectra available.**" (Lines 336–339)

**Response to the comments of Referee #2 on "Influence of anthropogenic pollution on the molecular composition of organic aerosols over a forest site in the Qinling Mountains region of central China"**

by Xin Zhang and Yuemei Han

We greatly thank Referee #2 for the helpful comments and suggestions on our manuscript (ID: egusphere-2025-519). We have seriously addressed all the issues raised by the referee and made relevant revisions on the manuscript. Please find below for the details of our responses to these comments. A copy of the manuscript and supplement with all the changes and other minor corrections tracked is also attached for reference.

(*The blue bold, green, and black fonts represent the Referee's comments, the related text in the manuscript, and the authors' responses, respectively.*)

**This manuscript presents observational evidence of the seasonal variability in organic aerosol composition. However, the research approach—combining seasonal sampling with high-resolution mass spectrometry—has been widely reported in previous studies. The study does not clearly provide novel mechanistic insights or substantial advancements in the understanding of anthropogenic–biogenic interactions. As such, the conclusions drawn lack originality and scientific novelty. I regret to conclude that the manuscript, in its current form, does not meet the publication standards of the journal. My main concerns are as follows:**

Response: We really appreciate the referee for providing these comments on our manuscript. We believe that it might be to some extent true that "combining seasonal sampling with high-resolution mass spectrometry has been widely reported in previous studies" in certain areas of the world. However, this is definitely not the case in the Qinling Mountains region of central China, despite its significant roles of serving as natural geographical and climatic boundary between the northern and southern China. Actually, based on our latest literature survey, the

molecular composition of organic aerosols in the Qinling Mountains region has rarely been investigated to date, especially utilizing the advanced HRMS techniques. This point has been stated in the manuscript as follows: "So far to our knowledge, this study **for the first time reports** the overall molecular characteristics of organic aerosols between contrasting seasons in the Qinling Mountains region based on high-resolution Orbitrap mass spectrometry." (Lines 76–78)

Moreover, the research approach of HRMS analysis remains to be one of the most advanced technologies in the field of atmospheric aerosol research currently. In this study, we focused primarily on the application of HRMS technique in the Qinling Mountains region for organic molecular characterization, rather than the development of this research approach itself. Therefore, the originality and novelty of this study should not be only judged by the research approach used herein, but more importantly by the research objectives and scientific findings. We acknowledge that it is somewhat challenging to directly elucidate mechanistic insights from aerosol sampling and offline HRMS analysis under complex atmospheric conditions, compared with those using real-time measurement techniques in laboratory studies that have the capability of capturing rapid changes in specific aerosol reaction chemistry. For this sake, we have included more mechanistic insights from previous literature to strengthen our conclusions, as addressed in detail in the response to the referee's comment #4.

Taken these facts together, the originality and scientific novelty of this current study include the following main aspects: 1) it first provided a comprehensive insight on the chemical characteristics of organic aerosols at the molecular level in the Qinling Mountains region, which has unique and significant geographical and climatic features; 2) it first demonstrated the substantial influence of anthropogenic pollution on the organic molecular composition in this region; 3) it advanced our knowledge on the anthropogenic and biogenic interactions as well as elucidated its seasonal variabilities and underlying influencing factors. As a result, we believe that this study would initiate and promote more atmospheric aerosol researches to further understand the anthropogenic perturbations on air quality, atmospheric chemistry, and associated environmental and climate impacts in the Qinling Mountains region, and also provide important references for similar regions worldwide.

*1. Line 305: this interpretation lacks strong causal linkage, as biogenic emissions—particularly monoterpene-derived secondary organic aerosols (SOA)—also predominantly fall within the C₆–C₁₁ range. Therefore, the contribution from biogenic precursors should not be overlooked when interpreting the molecular composition in this carbon number range.*

Related sentence: "In contrast, the CHO and CHON species in ESI+ mode and CHONS species in both modes had increased number fractions at approximately $C_{6-11}$ in winter, possibly resulted from the enhanced influence of anthropogenic pollution."

Response: The referee's viewpoint here is slightly different from what we intended to express from this statement. In fact, we did not mean to rule out the contribution from biogenic precursors at the $C_6$–$C_{11}$ range, as addressed in the context of the same paragraph: "Among them, the number and peak area fractions of organic species were particularly dominated by the $C_5$–$C_{11}$ compounds, which can be derived from both biogenic and anthropogenic sources." (Lines 299–301). Rather, the key point here is that the increased number fractions of those species in winter were contributed by the enhanced influence of anthropogenic pollution from the surrounding areas. It is likely that the original expression was not clear enough to be understood. Therefore, we have highlighted this point for clarity in the revised manuscript, as follows:

"In contrast, **prominent increases were observed for the number fractions of CHO and CHON species in ESI+ mode and CHONS species in both modes at approximately $C_{6-11}$ range during the winter period, probably resulted mainly from the enhanced influence of anthropogenic pollution from the surrounding areas, compared with those of the summer period**." (Lines 306–309)

*2. Lines 365-370: here only discuss the seasonal variation of molecules identified under the ESI− mode using VK diagrams, while omitting the analysis of VK characteristics of compounds detected in the ESI+ mode. This incomplete treatment results in a weak logical connection between the data presented and the conclusions drawn, leading to a lack of coherence in the overall interpretation. Given that ESI+ typically captures a distinct subset*

*of organic compounds—often including important nitrogen- and sulfur-containing species—its exclusion leaves a significant gap in the discussion. I recommend the authors include a comparative analysis of VK diagrams under both ionization modes to ensure a more comprehensive and balanced understanding of the seasonal dynamics of organic aerosol composition.*

Related sentence: "The anthropogenic and biogenic organic tracer species identified in ESI− mode were further analyzed in the van Krevelen diagram based on their elemental ratios of O/C and H/C, as shown in Fig. 8a. Except for those derived from glyoxal, isoprene, and some anthropogenic precursors with O/C > 0.5, other tracer species were primarily distributed in the regime with O/C ≤ 0.5. In addition, the anthropogenic tracer species widely dispersed at the regime with H/C from approximately 0.5 to 2.5, whereas the biogenic ones were mainly situated at the H/C ≥ 1.5 regime."

Response: Since the VK diagrams for organic tracer species in ESI+ mode showed quite similar seasonal variation trends as those in ESI− mode, together with that organic tracer species from more precursors were identified in the ESI− mode, we therefore only discussed the results of negative mode in the original manuscript. As suggested by the referee, we have provided the results of VK diagrams for both ESI− and ESI+ modes in order to gain a more comprehensive understanding on the seasonal dynamics of organic aerosol composition. Specifically, we have revised the original Figure 8 by including the VK diagram results and seasonal variation of organic species in both ESI− and ESI+ modes (see the revised Figure 8 below). We have also modified and added the relevant discussions in the revised manuscript, as follows:

"The anthropogenic and biogenic organic tracer species identified in **the ESI− and ESI+ modes** were further analyzed in the van Krevelen diagram based on their elemental ratios of O/C and H/C, as shown in Fig. 8a. **Similar variation patterns were generally observed between the two modes. Specifically,** except for those derived from glyoxal, isoprene, and some anthropogenic precursors with O/C > 0.5, other tracer species were primarily distributed in the regime with O/C ≤ 0.5." (Lines 365–368)

"Figure 8b presents the example results on two different days **(that is, August 30 and January 26) obtained in the ESI− mode.**" (Lines 375–376)

"**Similar tendencies were also found for those in the two modes** across the entire study periods in summer and winter (Fig. 8c)." (Lines 380–381)

"**In ESI− mode,** the number fraction of organic species in the aromatic regime was on average 21 ± 2% on individual days in winter, which was higher than those of 17 ± 2% in summer." (Lines 381–382)

"**Likewise, in ESI+ mode, the number and peak area fractions of aromatic species were slightly higher in winter than in summer, while those of aliphatic species were higher in summer.**" (Lines 384–386)

[Figure]

Figure 8. van Krevelen diagrams plotting of (a) organic tracer species derived from anthropogenic and biogenic precursors **in both modes (as listed in Table S1–S5)**, and (b) organic species identified on a summer day (August 30) and a winter day (January 26) in ESI− mode. (c) Averaged number and peak area fractions of organic species in aromatic and

aliphatic molecular structures for summer and winter samples **in ESI− and ESI+ modes**. In panels (a, b), the shaded pink (H/C ≥ 1.5, O/C ≤ 0.5) and cyan (H/C ≤ 1.0, O/C ≤ 0.5) areas represent aliphatic and aromatic regimes, respectively. The O/C was replaced by (O−3S)/C or (O−3S−2N)/C in the case of organosulfates or nitrooxy-organosulfates. The error bars in panel (c) represent 1σ standard deviations for all the samples in each season.

*3. Lines 395-410: The conclusions regarding the influence of relative humidity on the molecular characteristics of organic species are solely based on compounds identified under the ESI− mode. This raises concerns about the completeness of the analysis, as ESI− and ESI+ modes often detect different classes of organic compounds with distinct physicochemical properties. Relying only on ESI− data may lead to a biased or incomplete understanding of humidity-driven processes. A more balanced interpretation should incorporate results from both ionization modes to better capture the full range of organic species affected by relative humidity.*

Response: As suggested by the referee, we have performed further analysis with regard to the influence of relative humidity on the molecular characteristics of organic species in the ESI+ mode during the study periods. The results eventually showed that the overall variation trends of the number of organic species under different RH conditions were to some extent similar between the two ionization modes (see Figure S2 below and also in the revised Supplement), although the RH influence on organic species in the ESI+ mode were not as strong as those in the ESI− mode. This is most likely associated with the distinct physicochemical properties of organic species in these two ionization modes, that is, basic organic substances were readily protonated and thus detected in the positive mode, while acidic substances were easily deprotonated and more sensitive in the negative mode. As a result, we have added the relevant discussions regarding the influence of RH on organic species in the ESI+ mode in the revised manuscript, as follows:

"**In contrast, the similar but weaker dependence on RH was found for organic species in the ESI+ mode (Fig. S2 in the Supplement), probably associated in part with their distinct physicochemical properties compared with those in the ESI− mode. This result**

**also indicates that the influence of RH could be stronger for acidic organic substances, since these were more readily deprotonated and sensitive to be detected in negative mode than the basic ones (Laskin et al., 2018).**" (Lines 416–419)

The following literature also has been added in the References section of the revised manuscript:

**Laskin, J., Laskin, A., and Nizkorodov, S. A.: Mass spectrometry analysis in atmospheric chemistry, Anal. Chem., 90, 166–189, https://doi.org/10.1021/acs.analchem.7b04249, 2018.** (Lines 647–648)

[Figure]

**Figure S2. Species number and the relative fractions of organic molecular subgroups in positive ESI mode under varied relative humidity conditions in (a, b) summer and (c, d) winter periods. The symbols of each type shared the same RH values (that is, 62 ± 6%, 75 ± 4%, 93 ± 2% in summer and 43 ± 8%, 58 ± 7%, 88 ± 5% in winter) but were offset horizontally for a clear vision. The horizontal error bars in the CHO species represent 1σ standard deviations of RH values. The vertical error bars represent 1σ standard deviations for all the samples in each RH condition.**

In addition, we have also added the following result (Figure S4) and relevant discussions regarding the influence of temperature on organic species in ESI+ mode into the Supplement, for a more comprehensive understanding on the influence of meteorological factors.

"The species numbers of all organic subgroups **in ESI− mode** were almost comparable at the mean temperatures of 21.5 ± 0.8 and 24.5 ± 0.8 °C in summer, while they were largely increased at the highest temperature level of 27.6 ± 1.0 °C **(Fig. S3 in the Supplement**)." (Lines 430–432)

"**The species number and fraction of organic subgroups in ESI+ mode varied insignificantly with temperature as well (Fig. S4 in the Supplement).**" (Lines 433–434)

[Figure]

**Figure S4. Species number and the relative fractions of organic molecular subgroups in positive ESI mode under varied temperature conditions in (a) summer and (b) winter periods. The symbols of each type shared the same temperature values (that is, 21.5 ± 0.8, 24.5 ± 0.8, 27.6 ± 1.0 °C in summer and 0.3 ± 0.6, 2.0 ± 0.7, 3.8 ± 1.0 °C in winter) but were offset horizontally for a clear vision. The horizontal error bars in the CHO species represent 1σ standard deviations of temperature values. The vertical error bars represent 1σ standard deviations for all the samples under each temperature condition.**

*4. Lines 450-475: The authors primarily rely on positive and negative correlations with gaseous and particulate pollutants, as well as SOR, NOR, and Ox, to infer their influence on the chemical composition of organic aerosols. However, the correlation-based analysis lacks mechanistic support, making the conclusions less convincing. Furthermore, the discussion remains rather superficial and does not adequately elucidate the underlying interactions between pollutants and organic aerosol formation. It is recommended that the authors incorporate mechanistic insights from previous literature to strengthen the scientific basis and credibility of their conclusions.*

Response: We hope the referee could understand that the dataset of this study was mainly derived from HRMS measurement of aerosol filter samples, thereby the correlation-based analysis of organic species with air pollutants was one of the efficient methods to explore the underlying interactions. We acknowledge that although the offline HRMS analysis had advantages in determining organic molecular composition and identifying reaction products, it remains challenging to directly elucidate specific reaction mechanisms, especially under atmospheric conditions with complex influences of multiple environmental factors. Actually, the conclusions drawn from the correlation-based analysis in our current study are highly consistent with some previous laboratory studies. Therefore, as recommended by the referee, we have incorporated the following mechanistic insights to strengthen the scientific basis and credibility of our conclusions in the revised manuscript:

"This phenomenon **could partly** result from the chemical transformation of these species into **other chemically diverse substances (e.g., CHON and CHONS) through acid-catalyzed heterogeneous reactions and/or aqueous phase chemistry in the presence of anthropogenic air pollutants (Darer et al., 2011; Fan et al., 2022; Kwong et al., 2018; Lam et al., 2019).**" (Lines 466–469)

"**Another proposed explanation is that the productions of CHO and CHOS species were to some extent governed by the consumption of anthropogenic pollutants at the low concentration levels in summer (Fig. 2c), as those reported in some other ambient atmospheres (Lin et al., 2022; Riva et al., 2019; Wang et al., 2020).**" (Lines 469–472)

"**Possible reaction pathways include such as biogenic isoprene and monoterpene**

precursors being oxidized by nitrate radicals to form condensable organic nitrates and nitrooxy-organosulfates (Hamilton et al., 2021; Ng et al., 2017)." (Lines 473–475)

"These species might be partly associated with primary emissions (Song et al., 2018; Zhong et al., 2023), such as those of biomass burning and fossil fuel combustion from the surrounding rural and urban areas." (Lines 479–480)

"This is consistent with previous studies that oxidant levels were one of the most important governing factors for the production of organosulfur species by limiting the oxidation of organic precursors (Bryant et al., 2021; Wang et al., 2021f)." (Lines 487–489)

The following literatures have been added in the References section to support our conclusions in the revised manuscript:

[revised manuscript text omitted]

The references are listed as follows: [1] Surratt et al., 2008; [2] Lim et al., 2010; [3] Perri et al., 2010; [4] Shalamzari et al., 2016; [5] Bryant et al., 2023; [6] Schindelka et al., 2013; [7] Safi Shalamzari et al., 2013; [8] Wang et al., 2016; [9] Szmigielski, 2016; [10] Riva et al., 2016a; [11] Riva et al., 2016b; [12] Cai et al., 2020; [13] Chan et al., 2011.

**Table S2. Lists of organosulfate (including nitrooxy-organosulfate) compounds derived from anthropogenic sources and detected in negative ionization mode.**

| Precursor | Neutral mass | Compound | Reference | Precursor | Neutral mass | Compound | Reference |
|---|---|---|---|---|---|---|---|
| **Long-chain alkanes** | 210.2481 | $C_7H_{14}O_5S$ | 11, 12 | **Naphthalene** | 174.1744 | $C_6H_6O_4S$ | 12, 14 |
| | 196.2645 | $C_7H_{16}O_4S$ | 8, 11 | | 186.1851 | $C_7H_6O_4S$ | 12, 14 |
| | 210.2911 | $C_8H_{18}O_4S$ | 8, 11 | | 202.1845 | $C_7H_6O_5S$ | 8, 14 |
| | 238.3012 | $C_9H_{18}O_5S$ | 8, 11 | | 188.2010 | $C_7H_8O_4S$ | 8, 14 |
| | 270.3000 | $C_9H_{18}O_7S$ | 8, 11 | | 230.1946 | $C_8H_6O_6S$ | 8, 14 |
| | 286.2994 | $C_9H_{18}O_8S$ | 11, 12 | | 200.2117 | $C_8H_8O_4S$ | 8, 14 |
| | 256.3165 | $C_9H_{20}O_6S$ | 11, 12 | | 216.2111 | $C_8H_8O_5S$ | 8, 14 |
| | 296.2942 | $C_{10}H_{16}O_8S$ | 11, 12 | | 230.2377 | $C_9H_{10}O_5S$ | 8, 14 |
| | 312.2936 | $C_{10}H_{16}O_9S$ | 8, 11 | | 216.2542 | $C_9H_{12}O_4S$ | 12, 14 |
| | 252.3278 | $C_{10}H_{20}O_5S$ | 8, 11 | | 232.2536 | $C_9H_{12}O_5S$ | 15 |
| | 268.3272 | $C_{10}H_{20}O_6S$ | 8, 11 | | 228.2218 | $C_9H_8O_5S$ | 14 |
| | 300.3260 | $C_{10}H_{20}O_8S$ | 11, 12 | | 258.2478 | $C_{10}H_{10}O_6S$ | 8, 14 |
| | 308.3480 | $C_{12}H_{20}O_7S$ | 11, 12 | | 274.2472 | $C_{10}H_{10}O_7S$ | 8, 14 |
| | 280.3810 | $C_{12}H_{24}O_5S$ | 8, 11 | | 276.2631 | $C_{10}H_{12}O_7S$ | 8, 14 |
| | 266.3974 | $C_{12}H_{26}O_4S$ | 8, 11 | | 288.2738 | $C_{11}H_{12}O_7S$ | 8, 14 |
| **Diesel** | 138.1423 | $C_3H_6O_4S$ | 16 | | 290.2897 | $C_{11}H_{14}O_7S$ | 14 |
| | 152.1689 | $C_4H_8O_4S$ | 16 | | 219.1720 | $C_6H_5NO_6S$ | 14 |
| | 196.1784 | $C_5H_8O_6S$ | 16 | | 321.2606 | $C_{10}H_{11}NO_9S$ | 12, 14 |
| **Cyclohexene** | 166.1524 | $C_4H_6O_5S$ | 17 | | | | |
| | 210.2050 | $C_6H_{10}O_6S$ | 17 | | | | |

The references are listed as follows: [8] Wang et al., 2016; [11] Riva et al., 2016b; [12] Cai et al., 2020; [14] Riva et al., 2015a; [15] Wang et al., 2019; [16] Blair et al., 2017; [17] Liu et al., 2017.

**Table S3. Lists of CHO tracer species derived from biogenic/anthropogenic sources and detected in negative ionization mode.**

| Source | Natural mass | Compound | Potential identity/precursor | Reference |
|---|---|---|---|---|
| **Biogenic** | 118.0880 | $C_4H_6O_4$ | Succinic acid | 18 |
| | 172.1785 | $C_8H_{12}O_4$ | Terpenylic acid/α-pinene | 19 |
| | 188.1779 | $C_8H_{12}O_5$ | Terpenoic acid/α-pinene; β-pinene | 20 |
| | 204.1773 | $C_8H_{12}O_6$ | 3-methyl-1,2,3-butanetricarboxylic acid/α-pinene | 21 |
| | 174.1944 | $C_8H_{14}O_4$ | Suberic acid/unsaturated fatty acids | 22 |
| | 186.2051 | $C_9H_{14}O_4$ | Pinic acid/α-pinene; β-pinene | 23 |
| | 188.2209 | $C_9H_{16}O_4$ | Azelaic acid/unsaturated fatty acids | 22 |
| | 232.2304 | $C_{10}H_{16}O_6$ | Diaterpenylic acid acetate/α-pinene | 19 |
| | 272.4235 | $C_{16}H_{32}O_3$ | 3-hydroxyhexadecanoic acid/Gram-negative Bacteria | 24 |
| | 294.3859 | $C_{17}H_{26}O_4$ | Unknown/β-pinene | 25 |
| | 300.4766 | $C_{18}H_{36}O_3$ | 3-hydroxyoctadecanoic acid/Gram-negative Bacteria | 24 |
| **Anthropogenic** | 90.0779 | $C_3H_6O_3$ | Unknown/Bonfire | 26 |
| | 146.1412 | $C_6H_{10}O_4$ | Adipic acid/Cyclohexene | 27, 28 |
| | 160.1678 | $C_7H_{12}O_4$ | Pimelic acid/Cyclohexene | 27 |
| | 122.1213 | $C_7H_6O_2$ | Benzoic acid/PAHs | 28, 29 |
| | 166.1308 | $C_8H_6O_4$ | Phthalic acid/PAHs | 29 |
| | 270.4076 | $C_{16}H_{30}O_3$ | 2-Oxo-tetredecanoic acid/Cooking emissions; oils | 30 |

The references are listed as follows: [18] Xu and Zhang, 2012; [19] Claeys et al., 2009; [20] Gómez-González et al., 2012; [21] Szmigielski et al., 2007; [22] Mochida et al., 2003; [23] Ma et al., 2007; [24] Watson and Chow, 2018; [25] Slade et al., 2019; [26] Priestley et al., 2018; [27] Hansen et al., 2014; [28] Riva et al., 2015b; [29] Williams et al., 2010; [30] Qi et al., 2019.

**Table S4. Lists of CHO tracer species derived from biogenic sources and detected in positive ionization mode.**

| Natural mass | Compound | Potential precursor | Reference |
|---|---|---|---|
| 128.1690 | $C_7H_{12}O_2$ | limonene | 31 |
| 140.1797 | $C_8H_{12}O_2$ | β-pinene, limonene | 31, 32 |
| 156.1791 | $C_8H_{12}O_3$ | α,β-pinene, limonene, sabinene | 31, 32, 33 |
| 158.1950 | $C_8H_{14}O_3$ | α-pinene, limonene | 31, 33 |
| 172.1785 | $C_8H_{12}O_4$ | α,β-pinene, limonene, Δ3-carene, sabinene | 34, 35 |
| 154.2063 | $C_9H_{14}O_2$ | β-pinene, limonene, sabinene | 31, 32 |
| 156.2221 | $C_9H_{16}O_2$ | limonene | 31 |
| 168.1898 | $C_9H_{12}O_3$ | limonene | 31 |
| 170.2057 | $C_9H_{14}O_3$ | α,β-pinene, limonene, Δ3-carene, sabinene | 31, 33, 34 |
| 184.1892 | $C_9H_{12}O_4$ | α-pinene, limonene | 31, 33 |
| 186.2051 | $C_9H_{14}O_4$ | α,β-pinene, limonene, Δ3-carene, sabinene | 34, 36, 37 |
| 188.2209 | $C_9H_{16}O_4$ | α,β-pinene, limonene | 34, 38 |
| 168.2328 | $C_{10}H_{16}O_2$ | limonene | 31 |
| 180.2005 | $C_{10}H_{12}O_3$ | limonene | 31 |
| 182.2164 | $C_{10}H_{14}O_3$ | limonene | 31 |
| 184.2322 | $C_{10}H_{16}O_3$ | α,β-pinene, limonene, Δ3-carene | 34, 35 |
| 186.2481 | $C_{10}H_{18}O_3$ | α-pinene, limonene | 31, 33 |
| 198.2158 | $C_{10}H_{14}O_4$ | α-pinene, limonene | 31, 33 |
| 200.2316 | $C_{10}H_{16}O_4$ | α,β-pinene, limonene, sabinene | 31, 33 |
| 198.2588 | $C_{11}H_{18}O_3$ | limonene | 31 |
| 212.2423 | $C_{11}H_{16}O_4$ | α-pinene, limonene | 31, 33 |
| 214.2582 | $C_{11}H_{18}O_4$ | α-pinene, limonene | 31, 33 |
| 222.2372 | $C_{12}H_{14}O_4$ | limonene | 31 |
| 226.2689 | $C_{12}H_{18}O_4$ | α-pinene, limonene | 31, 33 |
| 240.2955 | $C_{13}H_{20}O_4$ | α-pinene, limonene | 31, 33 |
| 254.3221 | $C_{14}H_{22}O_4$ | limonene | 31 |
| 268.3487 | $C_{15}H_{24}O_4$ | α-pinene, limonene | 31, 33 |

The references are listed as follows: [31] Walser et al., 2008; [32] Yu et al., 1999; [33] Putman et al., 2012; [34] Yasmeen et al., 2011; [35] Kourtchev et al., 2014; [36] Glasius et al., 2000; [37] Camredon et al., 2010; [38] Kourtchev et al., 2015.

**Table S5. Lists of CHN tracer species derived from anthropogenic sources and detected in positive ionization mode.[a]**

| Natural mass | Compound | Natural mass | Compound | Natural mass | Compound |
|---|---|---|---|---|---|
| 98.1463 | $C_5H_{10}N_2$ | 130.1466 | $C_8H_6N_2$ | 155.1959 | $C_{11}H_9N$ |
| 100.1622 | $C_5H_{12}N_2$ | 145.1613 | $C_8H_7N_3$ | 183.2093 | $C_{11}H_9N_3$ |
| 94.1145 | $C_5H_6N_2$ | 132.1625 | $C_8H_8N_2$ | 169.2224 | $C_{12}H_{11}N$ |
| 96.1304 | $C_5H_8N_2$ | 147.1772 | $C_8H_9N_3$ | 197.2358 | $C_{12}H_{11}N_3$ |
| 111.1451 | $C_5H_9N_3$ | 146.1891 | $C_9H_{10}N_2$ | 184.2371 | $C_{12}H_{12}N_2$ |
| 110.1570 | $C_6H_{10}N_2$ | 161.2037 | $C_9H_{11}N_3$ | 199.2517 | $C_{12}H_{13}N_3$ |
| 112.1729 | $C_6H_{12}N_2$ | 148.2050 | $C_9H_{12}N_2$ | 186.2530 | $C_{12}H_{14}N_2$ |
| 114.1888 | $C_6H_{14}N_2$ | 157.1720 | $C_9H_7N_3$ | 201.2676 | $C_{12}H_{15}N_3$ |
| 119.1240 | $C_6H_5N_3$ | 144.1732 | $C_9H_8N_2$ | 203.2835 | $C_{12}H_{17}N_3$ |
| 108.1411 | $C_6H_8N_2$ | 159.1879 | $C_9H_9N_3$ | 182.2212 | $C_{12}H_{10}N_2$ |
| 123.1558 | $C_6H_9N_3$ | 158.1998 | $C_{10}H_{10}N_2$ | 180.2053 | $C_{12}H_8N_2$ |
| 122.1677 | $C_7H_{10}N_2$ | 173.2144 | $C_{10}H_{11}N_3$ | 194.2319 | $C_{13}H_{10}N_2$ |
| 137.1823 | $C_7H_{11}N_3$ | 160.2157 | $C_{10}H_{12}N_2$ | 181.2331 | $C_{13}H_{11}N$ |
| 124.1836 | $C_7H_{12}N_2$ | 175.2303 | $C_{10}H_{13}N_3$ | 183.2490 | $C_{13}H_{13}N$ |
| 126.1995 | $C_7H_{14}N_2$ | 169.1827 | $C_{10}H_7N_3$ | 211.2624 | $C_{13}H_{13}N_3$ |
| 128.2153 | $C_7H_{16}N_2$ | 156.1839 | $C_{10}H_8N_2$ | 200.2795 | $C_{13}H_{16}N_2$ |
| 118.1359 | $C_7H_6N_2$ | 171.1986 | $C_{10}H_9N_3$ | 217.3101 | $C_{13}H_{19}N_3$ |
| 133.1506 | $C_7H_7N_3$ | 170.2105 | $C_{11}H_{10}N_2$ | 196.2478 | $C_{13}H_{12}N_2$ |
| 120.1518 | $C_7H_8N_2$ | 185.2251 | $C_{11}H_{11}N_3$ | 179.2173 | $C_{13}H_9N$ |
| 148.1652 | $C_7H_8N_4$ | 172.2264 | $C_{11}H_{12}N_2$ | 223.2731 | $C_{14}H_{13}N_3$ |
| 135.1665 | $C_7H_9N_3$ | 187.2410 | $C_{11}H_{13}N_3$ | 208.2585 | $C_{14}H_{12}N_2$ |
| 162.1918 | $C_8H_{10}N_4$ | 189.2569 | $C_{11}H_{15}N_3$ | 237.2997 | $C_{15}H_{15}N_3$ |
| 149.1930 | $C_8H_{11}N_3$ | 168.1946 | $C_{11}H_8N_2$ | 205.2545 | $C_{15}H_{11}N$ |

[a] These CHN tracer species are mainly derived from anthropogenic sources such as combustion of fossil fuels and biomass according to the following literature: (Jiang et al., 2022; Laskin et al., 2009; Lin et al., 2012; Mao et al., 2022; Song et al., 2022; Wang et al., 2017).

[Figure]

**Figure S1.** Peak area fractions of organic subgroups sorted by the carbon atoms number in molecular formulas assigned in (a) negative and (b) positive ESI modes for summer and winter samples. The error bars represent 1σ standard deviations for all samples in each season.

[Figure]

Figure S2.

[Figure]

**Figure S2. Species number and the relative fractions of organic molecular subgroups in positive ESI mode under varied relative humidity conditions in (a, b) summer and (c, d) winter periods. The symbols of each type shared the same RH values (that is, 62 ± 6%, 75 ± 4%, 93 ± 2% in summer and 43 ± 8%, 58 ± 7%, 88 ± 5% in winter) but were offset horizontally for a clear vision. The horizontal error bars in the CHO species represent 1σ standard deviations of RH values. The vertical error bars represent 1σ standard deviations for all the samples in each RH condition.**

[Figure]

**Figure S3.** Species number and the relative fractions of organic molecular subgroups in negative ESI mode under varied temperature conditions in (a, ) summer and ( b) winter periods. The symbols of each type shared the same temperature values (that is, 21.5 ± 0.8, 24.5 ± 0.8, 27.6 ± 1.0 °C in summer and 0.3 ± 0.6, 2.0 ± 0.7, 3.8 ± 1.0 °C in winter) but were offset horizontally for a clear vision. The horizontal error bars in the CHO species represent 1σ standard deviations of temperature values. The vertical error bars represent 1σ standard deviations for all the samples under each temperature condition.

[Figure]

**Figure S4. Species number and the relative fractions of organic molecular subgroups in positive ESI mode under varied temperature conditions in (a) summer and (b) winter periods. The symbols of each type shared the same temperature values (that is, 21.5 ± 0.8, 24.5 ± 0.8, 27.6 ± 1.0 °C in summer and 0.3 ± 0.6, 2.0 ± 0.7, 3.8 ± 1.0 °C in winter) but were offset horizontally for a clear vision. The horizontal error bars in the CHO species represent 1σ standard deviations of temperature values. The vertical error bars represent 1σ standard deviations for all the samples under each temperature condition.**

---

## Author Response (AR2)

**Response to the Referee's comments on "Influence of anthropogenic pollution on the molecular composition of organic aerosols over a forest site in the Qinling Mountains region of central China"**

by Xin Zhang and Yuemei Han

We thank the referee #2 for taking time and providing further critical feedbacks on our manuscript (ID: egusphere-2025-519-R1). We have seriously taken into account of all the concerns raised by the referee and carefully revised the manuscript accordingly. Please find below our detailed point-to-point responses to all those concerns.

*I-1. This manuscript presents observational evidence of the seasonal variability in organic aerosol composition. However, it is important to note that the electrospray Ionization interface that couples the UPLC and Orbitrap MS is known for its varying ionization efficiencies across different compounds. The sensitivity of this technique can vary substantially by orders of magnitude, depending on the compound structure, functional group, and the characteristics of the sample matrix.*

Response: We thank the referee for pointing out the inherent limitation of ESI–HRMS technique regarding the ionization efficiency issue. However, in fact, this technique remains one of the most promising and powerful techniques in the research field of atmospheric and aerosol chemistry, despite some inherent drawbacks that need to be improve progressively. The claimed issue here itself could be applicable for any ESI–HRMS instruments, whereas cautions should be paid to utilize it for specific research topics. The ionization efficiency issue is in particular crucial for studies reporting the absolution mass concentrations of organic compounds. In our present study, however, the absolution mass concentrations of individual organic compounds were not a primary focus. Also, it is not possible to do this especially for a dataset derived from non-target screening of massive organic species measured by the HRMS, unless using some prediction approaches yet still with large uncertainties. Therefore, depending on the features of the dataset, our present study aims to

investigate the influence of anthropogenic pollution on organic aerosol composition based on the parallel comparison of variation trends in species number and peak area between different seasons. The relative changes in the seasonal variation patterns of species number and peak area did reflect the actually changes of organic species, rather than caused by the varied ionization efficiency.

*I-2. Consequently, the signal intensity in an ESI spectrum does not necessarily serve as a direct and reliable indicator of compound abundance. This variability poses challenges in accurately correlating ESI+ and ESI- ion abundances with the actual abundance of the compounds in a given sample.*

Response: It is obvious that we did not simply use the signal intensity in an ESI spectrum for analysis in the current study. Rather, by combining the results of UHPLC chromatogram with HRMS mass spectra, the organic species number and peak area were obtained for discussion, particularly in respect of their variation trends from parallel comparison of different samples. Also, for each individual organic compound, given that the ESI–Orbitrap MS operation parameters being controlled consistently for all the samples in our study, the obtained peak area should linearly correlate with their real abundance in the atmosphere. Therefore, the observed relative variations in the species number and peak area between different samples cannot be simply explained by the changing ionization efficiency.

*I-3. The high-resolution spectra offered by the Orbitrap MS only provide elemental composition information about the molecules. It is a major leap to connect this data to molecular identities, and discussions regarding functional groups or chemical details could be subjected to large uncertainties.*

Response: This statement apparently underestimates the powerful capabilities of HRMS and also ignores the tremendous efforts that have been made by the entire HRMS research community thus far. Based on the HRMS-derived exactive mass and elemental composition of organic molecules, along with other parameters such as chromatogram retention time and

peak area, a variety of reliable analytical methods and algorithms have been well-established in the past decades, those including but not limited to: elemental ratios, carbon oxidation states, ring and double-bond equivalent, aromaticity equivalent, maximum carbonyl ratio, van Krevelen diagrams, Kendrick mass defect, and so on (e.g., Nizkorodov et al., 2011; Nozière et al., 2015; Johnston and Kerecman, 2019; Zhang et al., 2023; Lin et al., 2025). These methods and relevant analyses have already provided extensive valuable information for understanding the detailed molecular identities and characteristics of organic species in the atmospheric and aerosol research field.

*References:*

Nizkorodov, S. A., Laskin, J., and Laskin, A.: Molecular chemistry of organic aerosols through the application of high resolution mass spectrometry, Phys. Chem. Chem. Phys., 13, 3612–3629, https://doi.org/10.1039/c0cp02032j, 2011.

Nozière, B., Kalberer, M., Claeys, M., Allan, J., D'Anna, B., Decesari, S., Finessi, E., Glasius, M., Grgic, I., Hamilton, J. F., et al.: The molecular identification of organic compounds in the atmosphere: state of the art and challenges, Chem. Rev., 115, 3919–3983, https://doi.org/10.1021/cr5003485, 2015.

Johnston, M. V and Kerecman, D. E.: Molecular characterization of atmospheric organic aerosol by mass spectrometry, Annu. Rev. Phys. Chem., 12, 247–274, https://doi.org/10.1146/annurev-anchem-061516-045135, 2019.

Zhang, W., Xu, L., and Zhang, H.: Recent advances in mass spectrometry techniques for atmospheric chemistry research on molecular-level, Mass. Spectrom. Rev., https://doi.org/10.1002/mas.21857, 2023.

Lin, Y., Zhang, X., Li, L., Li, Z., Wang, R., Xing, S., and Han, Y.: Review on the molecular characterization of atmospheric organic aerosols using high-resolution Orbitrap mass spectrometry: Techniques, applications, and perspectives, Aerosol Sci. Eng., https://doi.org/10.1007/s41810-025-00332-1, 2025.

*I-4. Overall, I perceive this manuscript as lacking of scientific rigor, with conclusions and findings that are short of specificity and novelty.*

Response: We disagree with this viewpoint. Given that all the raised concerns are focused on the instrumental technical and methodology aspects, while those are not really applicable to our present study, it has no scientific basis and irrational to claim that our study are short of specificity and novelty. As we clearly addressed in the 1st round of response, the specificity and novelty of this study rely primarily on the unique scientific question and findings in the studied Qinling Mountains region with significant geographical and climatic features, rather than to develop technique or methodology itself.

The specificity and novelty of this study are summarized as follows: (1) It first provided a comprehensive insight on the chemical characteristics of organic aerosols at the molecular level in the Qinling Mountains region, which has unique and significant geographical and climatic features; (2) It first demonstrated the substantial influence of anthropogenic pollution on the organic molecular composition in this region; (3) It advanced our knowledge on the anthropogenic and biogenic interactions as well as elucidated its seasonal variabilities and underlying influencing factors. We have revised the following statement to further highlight the novelty of this study:

"So far to our knowledge, this study for the first time reports the overall molecular characteristics of organic aerosols between contrasting seasons **and provides direct evidence for the large influence of anthropogenic pollution** in the Qinling Mountains region based on high-resolution Orbitrap mass spectrometry." (Lines 76–78)

In addition, we also noticed that the referee's evaluation on "Scientific significance" of this study was even degraded in this 2nd round, comparing with those in the 1st round of review process, although we kept on improving our manuscript according to the referees' comments. We would be grateful if this study could be evaluated more objectively.

*II-1. This study employs a characterization approach based on exact mass formulae in combination with literature references. While high-resolution mass spectrometry offers clear advantages in molecular formula determination, its application to a highly complex mixture such as that investigated here—comprising a large number of compounds with*

*varying degrees of oxidation—presents significant challenges and uncertainties. These include, but are not limited to, mass spectral data processing strategies (e.g., background and noise subtraction), adduct formation and identification, charge state determination, accurate mass measurements, elemental composition assignments, and complex data-dependent acquisition setups.*

Response: We thank the referee for highlighting the inherent challenges and uncertainties associated with the molecular characterization of highly complex mixtures by HRMS. The raised concerns here would influence the data interpretation in certain cases. However, by implementing the rigorous procedures for data processing in this study, we are confident that the presented results are robust and reliably represents the chemical complexity of organic species in all the samples. The following explanations are specified to address each of these points:

*a. Mass spectral data processing strategies:* We utilized the MZmine software with carefully optimized parameter settings for the processing of raw data from UHPLC–HRMS analysis. This software has been well-developed and widely used in the HRMS research community, offering core MS data processing procedures such as peak detection, shoulder peak filtering, chromatogram builder and deconvolution, deisotoping, adducts and peak complex searching, identification, duplicate peak filtering, and other functions (Pluskal et al., 2010; Schmid et al., 2023). On the other hand, we have established reliable parameters and procedures based on a number of previous studies in our research group (e.g., Lin et al., 2022; Han et al., 2023; Li et al., 2024; Zhang et al., 2025) and also further optimized these parameters for the current study. We have also added a relevant statement on the background and noise subtraction for aerosol samples in the revised manuscript, as follows:

"**The field blanks were also extracted and chemically analyzed using the same procedures, the results of which were used to correct any potential artifacts and backgrounds for the aerosol samples.**" (Lines 122–123)

*b. Adduct formation and identification:* The adduct annotation has already been taken into account for the raw mass spectral analysis using the MZmine software. We have constrained

the formula assignments by specifically searching for common adducts such as protonated, deprotonated, and other ions (e.g., $[M + H]^+$ and $[M + Na]^+$ in the positive ESI mode; $[M - H]^-$ in the negative ESI mode) to minimize ambiguity.

*c. Charge state determination:* The ESI source used in our study primarily produced single charged ions, which was confirmed by isotopic pattern matching in the MZmine software analysis procedures. This is also very common for the ESI sources producing single charged ions, in particular for atmospheric organic aerosol composition in the studied mass ranges at *m/z* 50–750.

*d. Accurate mass measurements:* As stated in our manuscript, the Orbitrap HRMS instrument was externally calibrated using Thermo Scientific Pierce standard calibration solutions before the measurement. Also, all the molecular formula assignments in this study were performed using an accurate mass tolerance of 2 ppm. Therefore, these procedures have guaranteed the high reliability of the mass accuracy measurement.

*e. Elemental composition assignments:* The organic molecular formulae were constrained to containing C, H, O, N, S elements with a mass tolerance of 2 ppm in this study, following the commonly acceptable HRMS constraints for atmospheric organic aerosols. Also, the assigned molecular species were further screened using elemental ratios of H/C (0.3–3.0), O/C (0–3), N/C (0–1.3), and S/C (0–0.8) to ensure their presence in nature. The formulas disobeying the nitrogen rule for even electron ions were also excluded from the assignment lists. These procedures ensure a high degree of precision in the assignment of elemental composition in the current study.

*f. Complex data-dependent acquisition setups:* In our current study, we did not employ the traditional data-dependent acquisition setup in the HRMS measurement. Rather, in order to capture the overall profiles of organic aerosols, we employed a non-target screening in full MS scan mode without fragmentation. As stated in the manuscript: "The mass range was set at m/z 50–750 **in full MS scan mode**, with the mass resolution of approximately 140,000 at m/z 200." (Lines 113–114). We have also carefully tuned the experimental conditions and acquisition parameters to ensure the reproducibility and minimal bias during the HRMS analysis.

*References:*

Pluskal, T., Castillo, S., Villar-Briones, A., and Oresic, M.: MZmine 2: Modular framework for processing, visualizing, and analyzing mass spectrometry-based molecular profile data, BMC Bioinf., 11, https://doi.org/10.1186/1471-2105-11-395, 2010.

Schmid, R., Heuckeroth, S., Korf, A., Smirnov, A., Myers, O., Dyrlund, T. S., Bushuiev, R., Murray, K. J., Hoffmann, N., Lu, M., et al.: Integrative analysis of multimodal mass spectrometry data in MZmine 3, Nat. Biotechnol., 41, 447–449, https://doi.org/10.1038/s41587-023-01690-2, 2023.

Lin, Y., Han, Y., Li, G., Wang, Q., Zhang, X., Li, Z., Li, L., Prévôt, A. S. H., and Cao, J.: Molecular characteristics of atmospheric organosulfates during summer and winter seasons in two cities of southern and northern China, J. Geophys. Res.-Atmos., 127, https://doi.org/10.1029/2022jd036672, 2022.

Han, Y., Zhang, X., Li, L., Lin, Y., Zhu, C., Zhang, N., Wang, Q., and Cao, J.: Enhanced production of organosulfur species during a severe winter haze episode in the Guanzhong basin of northwest China, Environ. Sci. Technol., 57, 8708–8718, https://doi.org/10.1021/acs.est.3c02914, 2023.

Li, L., Han, Y., Li, J., Lin, Y., Zhang, X., Wang, Q., and Cao, J.: Effects of photochemical aging on the molecular composition of organic aerosols derived from agricultural biomass burning in whole combustion process, Sci. Total Environ., 946, https://doi.org/10.1016/j.scitotenv.2024.174152, 2024.

Zhang, X., Li, L., Lin, Y., Wang, R., Zhu, C., Xiao, S., Cao, J., and Han, Y.: Chemical diversity of organosulfur species in various atmospheric environments over the Guanzhong basin of northwest China, J. Geophys. Res.-Atmos., 130, https://doi.org/10.1029/2024jd042478, 2025.

*II-2. In the present manuscript, structural elucidation relies solely on high-resolution mass spectra without MS/MS (including MS² and MS³) validation for standards and key compounds in field samples, and without the provision of chromatograms, mass spectra, or MS/MS spectra for the main tracer species. These omissions limit the robustness and verifiability of the reported conclusions.*

Response: It might be true in terms that the proposed MS/MS validation for standards and specific compounds would further strengthen the molecular structural elucidation of organic aerosols. In the current study, however, the extensive datasets were obtained from non-target screening in full MS scan mode using the HRMS. The non-target screening analysis herein allows for a more comprehensive investigation of massive organic species, rather than target screening analysis only for a limited number of specific compounds. Here the molecular structural elucidation is mainly related to the results regarding the aliphatics and aromatics in Figure 8 and relevant discussions in section 3.3. As summarized in Tables S1–S5 of the Supplement, hundreds of tracer species were identified in this study and also reported in literatures. The molecular structures of organic species were deduced according to the commonly reported aliphatic and aromatic regimes within the van Krevelen diagrams. In fact, the distributions of known tracer species did agree well with the two regimes, as seen in Figure 8a. Therefore, the method used for molecular formular elucidation in this study is different from those assumed by the referee, while the latter one is unrealistic for hundreds of organic compounds based on non-target screening analysis. Nevertheless, we have provided the chromatogram retention time to further identify each of those tracer species from the UHPLC–HRMS analysis, as presented in Tables S1–S5 of the Supplement.

*II-3. Addressing these concerns would require substantial additional experimental work, including targeted MS/MS validation and supplementary analyses, which appear to go beyond the scope of a standard revision. In light of this, I believe the manuscript, in its current form, is not yet suitable for publication in the journal.*

Response: As addressed above, the data processing strategies for the UHPLC–HRMS dataset were rigorously performed in our study, and the structural elucidation relies primarily on the commonly used approach, van Krevelen diagram analysis, which is more suitable for the non-target screening of massive organic molecules detected in full MS scan mode. While the referee proposed targeted MS/MS would to some extent further strengthen the structural elucidation, our analyses were constrained by stringent mass accuracy, isotopic pattern

matching, and cross-validation with previous studies, which significantly enhances the reliability of the results.

Meanwhile, we hope the referee can understand that scientific research should be essentially proceeded step by step. It is not possible for a single study to cover multiple topics, instead, each study should have its own primary focus and research emphasis. In our current study, the main objective was to provide a comprehensive molecular-level overview of organic aerosol profiles and in-depth understand the influence of anthropogenic pollution on their chemical composition in the Qinling Mountains region, for which the non-target screening with HRMS is a powerful and well-adopted approach. Considering its novel findings and significance in the unique Qinling region, along with the rigorous procedures for data processing and careful interpretation, we believe that this study provides sufficient values for publication in ACP journal.

*1. Majority of discussions in this work are based on the variation or comparison of different compound categories, while the ionization efficiency of different organic compounds varied a lot. I wonder if the authors consider or evaluate the influence of varied ionization efficiency of different compounds. How much the variations of molecular characterization across different seasons are caused by the changing of ionization efficiency?*

Response: We believe that the referee's viewpoints here might have overlooked the fact that we did not report the absolute mass concentrations in the present study. Since we seriously understand the inherent technical limitation of ESI with the varied ionization efficiency (IE) for individual organic compounds, we did not directly refer the observed results and relevant discussions to their absolute mass concentrations throughout the entire study. There are some previous studies tried predicting the IE of different organic compounds from ESI-HRMS measurement (e.g., Bieber et al., 2023; Evans et al., 2024; Wang et al., 2025), however, the reported methods still have many uncertainties and further optimizations are required to improve their robustness in practical applications. Give that this is still an open question and

no consensus conclusion reached in the entire HRMS research community so far, especially for studies using non-target screening methods, we therefore primarily focused on the variation patterns and comparison of organic species in terms of their species number and peak area in the present study.

Since the ionization efficiency was not applied to estimate the absolute mass concentration of individual compounds in our current study, the observed seasonal variations between summer and winter were exclusively caused by the actual changes of organic species, rather than the changing of ionization efficiency. Furthermore, the ESI-Orbitrap MS operation parameters in our study were controlled consistently during the measurement procedures for all samples, the reported peak area for individual organic compound was not likely affected substantially by the changing of ionization efficiency. Therefore, based on the parallel comparison of results derived from all the samples in the same measurement conditions, the seasonal changes of organic species can be well captured from this analysis.

*References:*

Bieber, S., Letzel, T., and Kruve, A.: Electrospray ionization efficiency predictions and analytical standard free quantification for SFC/ESI/HRMS, J. Am. Soc. Mass Spectrom., 34, 1511–1518, https://doi.org/10.1021/jasms.3c00156, 2023.

Evans, R. L., Bryant, D. J., Voliotis, A., Hu, D., Wu, H., Syafira, S. A., Oghama, O. E., McFiggans, G., Hamilton, J. F., and Rickard, A. R.: A semi-quantitative approach to nontarget compositional analysis of complex samples, Anal. Chem., 96, 18349–18358, https://doi.org/10.1021/acs.analchem.4c00819, 2024.

Wang, W. C., Amini, N., Huber, C., Kull, M., and Kruve, A.: Active learning improves ionization efficiency predictions and quantification in nontargeted LC/HRMS, Anal. Chem., 97, 13131–13139, https://doi.org/10.1021/acs.analchem.5c00816, 2025.

*2. Line 375-380: In Figure 8b, the authors present example results for two days—August 30 (summer) and January 26 (winter)—in ESI- mode. However, the rationale for selecting these specific days is not clearly explained. Are these days representative in terms of meteorological conditions, pollution levels, or biogenic emissions? Without a clear*

*justification, it is difficult to assess whether the observed seasonal differences can be generalized. The authors should clarify the criteria used for day selection and explain how these examples reflect the broader trends discussed in Figure 8c.*

Response: We could not agree with the referee regarding this point. The reason is explained as follows. Here the analyses of two individual days in Figure 8b is mainly used to establish the analytical method, whereas, more importantly, the final results based on this analysis for all the days across the entire study periods in both negative and positive ESI modes are summarized and presented in Figure 8c. Technically, there was (and should be) no specific criteria used for selecting these two example days. Give that it is impossible to present a massive dataset for all the individual days in a single graph, we therefore only exemplified the results of these two days on August 30 and January 26 for illustration purpose herein. Regardless of which days were selected to be present in Figure 8b, this will not affect the final results and conclusions drawn from the Figure 8c for the entire periods. Actually, the results of averaged species number and peak area fractions with error bars for all the samples across the entire study periods in Figure 8c did clearly reveal that the observed seasonal differences are generalized and reflect the broader trends between the two summer and winter seasons.

Therefore, to avoid any possible ambiguous understanding, we have revised and added the following statements in the main text to highlight the key points of Figure 8b and 8c:
"Figure 8b presents the example results **obtained in the ESI− mode** on two different days (that is, August 30 and January 26) **for illustration**." (Lines 375–376)
"**A statistical analysis was further performed for all the summer and winter samples detected in both ESI− and ESI+ modes across the entire study periods, as shown in Figure 8c, from which similar seasonal variation tendencies were also obtained as those of the above two example days.**" (Lines 380–382)

*3. Lines 507–512 and Lines 336–339: The comparison of tracer species and the identification of clear indicators for organosulfates and nitrooxy organosulfates are noted.*

*However, in HR-MS methods as the one used in this study, several structures proposed by the software can be associated with each formula and confusing and challenging decisions should be made for structural elucidation when hundreds if not thousands of compounds are present in the sample. Although these techniques can provide important information, I believe challenges are still presents when using these analytical techniques and special care should be used when interpreting data and elucidating structure from HR mass spectra. Therefore, additional analysis (e.g. MS/MS) should be conducted!*

Response: Since our extraction experiments and the UHPLC–HRMS analysis of aerosol filter samples in this study were performed around two years ago, there are currently no samples from the same campaign available for the MS/MS analysis. Nevertheless, in order to clarify the referee's concern here, we have performed another control experiment recently using the aerosol samples collected at the same site but in the second year, extracted the organic compounds using the same experimental procedures, and then analyzed the samples with the UHPLC–HRMS in both full MS mode and targeted MS/MS mode. An example result of the chromogram, MS1 spectrum, MS2 spectrum for one of the organosulfate compounds analyzed this time are presented below. Overall, for all the ten samples analyzed this time, the results showed good agreements between the assigned organosulfate (including nitrooxy-organosulfate) compounds from the accurate mass analysis and those detected from by the MS2 analysis. Specifically, a number of 665–779 organic species possessed the $-OSO_3H$ groups in the molecules for individual samples, which were roughly defined as organosulfates; within them, 560–570 of which were eventually further confirmed as organosulfates (such as with $HSO_4^-$, $SO_4^-$, and $SO_3^-$ ion fragments) based on their MS2 analysis, accounting for 73%–85% of those proposed from the structural elucidation. This percentage should be even higher for the dataset used in our current study, because only the key tracer species were used for analysis and discussion in the manuscript. Therefore, based on the control experiment, we believe that the following statement regarding the structural elucidation method and relevant discussions are certainly reliable:

"Among these, organosulfate and nitrooxy-organosulfate compounds (abbreviated as OSs hereafter), roughly defined as those with sufficient oxygen atoms to assign the $-OSO_3H$ and

−ONO$_2$ groups in the molecules (Brüggemann et al., 2020; Lin et al., 2012b), were important components in both seasons. Note that the designated OSs here would correspond to those at the upper limit, since no further fragmentation spectra **were analyzed herein**." (Lines 336–339)

[Figure]

Figure. An example result of the chromogram, MS1 spectrum, and MS2 spectrum for an organosulfate compound, C$_{10}$H$_{17}$NO$_7$S, derived from the UHPLC–HRMS analysis in the control experiment.

---

## Author Response (AR3)

Response to the Referee's comments on "Influence of anthropogenic pollution on the molecular composition of organic aerosols over a forest site in the Qinling Mountains region of central China"

by Xin Zhang and Yuemei Han

We greatly thank editor for the message and the valuable and insightful comments from referee on our manuscript (ID: egusphere-2025-519-R2). We have carefully revised the manuscript by considering all the concerns raised by the referee. Please find below our detailed point-to-point responses to each of these comments.

(The blue bold, green, and black fonts represent the Referee's comments, the related text in the manuscript, and the authors' responses, respectively.)

This study describes a detailed molecular characterization of PM2.5 sampled from a forest site in the Qinling Mountains, China. Ambient filter samples collected during summer and winter were analyzed by ultrahigh-performance liquid chromatography and high-resolution Orbitrap mass spectrometry followed by a non-target analysis with a focus on the seasonal variability of the organic aerosol composition. Furthermore, air quality measurements, meteorological data, and back-trajectory analyses were integrated to draw detailed conclusions on aerosol origin and chemical transformation pathways. The authors conclude that chemical composition variation indicates a stronger influence from biogenic precursors in summer, contrasting with a more diverse, largely anthropogenic influenced composition in winter.

Combining non-target screening with filter samples from different seasons to investigate differences in organic aerosol composition is a widely used approach leading to already known basic results regarding anthropogenic—biogenic interactions. Orbitrap MS data provides exact mass of measured compounds from which molecular formulae can be derived using a software-based approach. To further confirm the identity of tracer compounds unfortunately only a literature review was carried out while a combination with fragmentation spectra (MS2) and retention time comparison with well-known biogenic and anthropogenic standards would

have provided a significantly greater certainty regarding compound identification.

Additionally, using electrospray ionization introduces ionization efficiency variations for compounds of different classes and chemical functionalities preventing from direct correlation of compound abundance by comparing measured signal intensities, especially if both polarity modes (ESI- and ESI+) are used.

Although the study has certain methodological weaknesses and does not provide any new fundamental scientific insights, it nevertheless provides a certain overview of the chemical composition of PM2.5 aerosols at Qinling Mountain Station, which contributes to the general understanding of this region and encourages further studies. The manuscript is well written and structured with interesting illustrations that support the results. In general, this study is publishable and fits the thematic focus of ACP and could either be published as Research article or also maybe as Measurement report after addressing some minor comments:

Response: We greatly appreciate the referee for providing insightful and thoughtful comments on our manuscript. We have seriously addressed all the concerns and made relevant corrections in the revised manuscript according to these comments. Our detailed responses are described as follows.

**Comments**

1. L66: Can you define and describe this specific area (anthropogenic-biogenic intersection zones) more precisely? Why is this zone of special interest regarding aerosol chemical composition?

Related sentence: "However, the interactions of anthropogenic pollution with biogenic organic aerosols and the potential impacts are still rarely reported at the lower altitudes of this region to date, especially at the anthropogenic–biogenic intersection zones."

Response: We thank the referee for raising this question. As shown in the map of Figure 1, the sampling site in the present study was situated in the intersection area of Qinling Mountains region and Guanzhong basin, which were abundance of biogenic emissions and anthropogenic pollutants, respectively. Therefore, we defined this area to be the anthropogenic–biogenic

intersection zones. This zone is of special interest regarding aerosol chemical composition because it can reflect the anthropogenic perturbations on air quality, atmospheric chemistry, and associated climate impacts in the Qinling Mountains region. Therefore, we have added the following two statements in the revised manuscript, as follows:

"Since anthropogenic pollutants and biogenic emissions were prevalent in this area, it is thus considered as an anthropogenic-biogenic intersection zone." (Lines 88–89) "Understand these interactions will be valuable to elucidate the anthropogenic perturbations on air quality, atmospheric chemistry, and associated climate impacts in the Qinling Mountains region. (Lines 66–68)

2. Figure 1. (b, c): Can you include the origin of the mass concentrations in the caption. On which type of measurements is this data based?

Response: Yes, the origin of the organic mass concentrations have been included into the caption of Figure 1 in the revised manuscript, as follows:

"Figure 1. (a) A map showing the sampling site at northern foothill of the Qinling Mountains region in central China. (b, c) Back trajectories of air masses arrived at 500 m above the ground level over the sampling site and their cluster analysis (represented by the thick solid lines with percentages and numbers) during the summer and winter periods. The bottom left panels in (b, c) present the mean mass concentrations of organic matter that were measured using the carbon analyzer for aerosol samples collected on the days with the corresponding air mass directions." (Lines 98–102)

3. L101: It is rather unusual to provide extraction solvent related information in a weight mass unit. Could you translate here mass into volume?

Related sentence: "A quarter of each sample in 12.56 cm2 area was ultrasonically extracted using 9 g acetonitrile and water mixture in 9:1 volume for 30 min (3 g for 10 min, repeated three times)."

Response: Since the extraction solvent for each sample was weighted using an electronic

microbalance in our experimental procedures, we believe that reporting the weight mass unit would be more accurate to reflect the actual situation in this study. It is possible to transfer the mass into volume unit by assuming a density for the solvent mixture of acetonitrile and water, however, this would result in more uncertainties. Therefore, we decided to keep using the weight mass unit here eventually.

4. L102: Ultrasonication is no optimal method for filter extraction as it is known that the harsh conditions in the liquid phase due to free radical formation can lead to substantial changes in the chemical composition. Although cooling might reduce the impact, it will also reduce the extraction efficiency of the solvent mixture and therefore influence the extracted compound composition. Did you compare filter extraction by ultrasonication and orbital shaking for your specific samples?

Literature: Riesz P, Berdahl D, Christman CL. Free radical generation by ultrasound in aqueous and nonaqueous solutions. Environ Health Perspect. 1985 Dec; 64:233-52. doi: 10.1289/ehp.8564233

Miljevic, B., Hedayat, F., Stevanovic, S., Fairfull-Smith, K. E., Bottle, S. E., & Ristovski, Z. D. (2014). To Sonicate or Not to Sonicate PM Filters: Reactive Oxygen Species Generation Upon Ultrasonic Irradiation. Aerosol Science and Technology, 48(12), 1276–1284. https://doi.org/10.1080/02786826.2014.981330

Related sentence: "A quarter of each sample in 12.56 cm2 area was ultrasonically extracted using 9 g acetonitrile and water mixture in 9:1 volume for 30 min (3 g for 10 min, repeated three times). The extraction system was placed in a water–ice bath to eliminate potential evaporation or chemical reactions of aerosol components."

Response: In the present study, we did not directly compare the filter extraction between ultrasonication and orbital shaking for specific samples. However, this topic has been investigated previously in the following literature. According to Hettiyadura et al. (2015), for higher percentage of organic solvents (that is, acetonitrile and ultra-pure water in 95 : 5 by volume in their case), there were no degradation effects and only minor chemical differences observed between the ultrasonication and rotary shaking procedures for sample treatment. In fact, the ultrasonication was determined to be a better method due to its higher precision compared

with rotary shaking (Hettiyadura et al., 2015). Similarly, we used the solvent of acetonitrile and pure water mixture in 9:1 volume in our present study, thereby the chemical effects should be at least not significant.

Moreover, we performed the ultrasonic extraction of filter samples in a water—ice bath to reduce the temperature, the extraction procedure lasted 10 min for each time, and the same procedure was repeated three times for each sample. Therefore, we believe that the aerosol chemical composition and extraction efficiency should be insignificantly affected by the ultrasonication procedure in this study. We will also consider to design a set of experiments to thoroughly illustrate this issue in our future study.

**Reference:**

Hettiyadura, A. P. S., Stone, E. A., Kundu, S., Baker, Z., Geddes, E., Richards, K., and Humphry, T.: Determination of atmospheric organosulfates using HILIC chromatography with MS detection, Atmos. Meas. Tech., 8, 2347–2358, 2015.

5. L118: Why not include phosphorus as element in the data processing? Anthropogenic compounds such as flame retardants often include phosphate groups (e.g. Tricresyl Phosphate). Chlorine may also be worth including due to its use in pesticides, provided this is relevant to the sampling region. Could you rerun the MZmine data processing including the mentioned elements and give some feedback if you can see relevant compounds, especially when the wind is coming from the city region.

Related sentence: "Briefly, the elemental composition of organic molecular species was constrained to  $C_{1-40}H_{1-80}O_{0-50}N_{0-4}S_{0-2}$  in the two ionization modes, using a mass tolerance of 2 ppm."

Response: We included the C, H, O, N, and S elements in the data processing herein, because they are commonly reported components in aerosol particles. As suggested by the referee, we have rerun the MZmine data processing by adding the phosphorus (P, 0–2 atoms) and chlorine (Cl, 0–2 atoms) elements for molecular formula assignments. Indeed, the result is consistent well with our assumption that the organic species containing P and Cl elements were generally minor components, as shown in the figure below. Specifically, the summed number and peak area intensity of organic species containing P and Cl merely accounted for less than 6% and 2% of

those in total, respectively. Therefore, the main conclusions drawn from this study were not affected by the presence of those minor P- and Cl-containing species. Nevertheless, we agree that the P- and Cl-containing species would be an important topic warranting further investigation especially in some atmospheric environments such as those dominated by industrial emission sources.

Figure. (a–d) Species number and peak area intensity of organic molecular composition, along with their corresponding number and peak area fractions (e–h), obtained from the UHPLC–HRMS analysis in negative and positive ESI modes over the study periods. Here the elements of C, H, O, N, S, P, and Cl were included for the organic molecular formula assignments.

6. L129: This method for determining the PM mass concentration is highly uncertain. If possible mass concentrations should be based on measurement data like measured particle number size distributions (PNSD).

Related sentence: "Each filter sample was weighted using an electronic microbalance before and after sampling, the difference of which dividing by the sampled air volume was used for calculating PM2.5 mass concentration."

Response: As our present study was primarily based on the aerosol filter sampling and chemical analysis, the mass concentrations of PM2.5 on the filter samples were therefore obtained using the electronic microbalance weighting method. This is also the traditional and widely used method for PM mass analysis for aerosol filter samples. Given that we performed the weighing procedures under the temperature and relative humidity precisely controlled conditions to avoid potential biases, the obtained PM mass concentrations should be highly reliable. The online measurement of particle number—size distributions was not available in this study, which will be a good direction for our future studies. Therefore, we have added the following statement to address this point in the revised manuscript:

"Each filter sample was weighted before and after sampling using an electronic microbalance under well-controlled temperature ( $20 \pm 2$  °C) and relative humidity ( $30 \pm 2$ %) conditions, the difference of which dividing by the sampled air volume was used for calculating PM2.5 mass concentration." (Lines 134–136)

**7. L257: You should substantiate this statement by mentioning known biogenic emission tracers as MBTCA and pinic acid in the text if found in the data.**

Related sentence: "Since air masses from the surrounding Qinling Mountains area in the south were more prevalent at the sampling site in summer season (Fig. 1b), biogenic emissions from forest vegetation (especially those with larger molecules such as monoterpenes and sesquiterpenes) and their oxidation products might be key factors resulting in the higher organic molecular weight."

Response: These organic tracer compounds were indeed found from the sample analysis in this study, as listed in Table S1 and S3. Therefore, we have added this point in the revised manuscript, as follows:

"Since air masses from the surrounding Qinling Mountains area in the south were more prevalent at the sampling site in summer season (Fig. 1b), biogenic emissions from forest vegetation (especially those with larger molecules such as monoterpenes and sesquiterpenes) and their oxidation products might be key factors resulting in the higher organic molecular weight, as evidenced by the presence of a number of biogenic tracer species such as MBTCA

8. Overall, to address the shortcomings of the compound identification (no measured standards and no analyzed fragmentation patterns) you should include the compound identification level system of Schymanski et. al. 2014 stating the level of identification confidence in the manuscript.

Literature: Identifying Small Molecules via High Resolution Mass Spectrometry:

Communicating Confidence Emma L. Schymanski, Junho Jeon, Rebekka Gulde, Kathrin

Fenner, Matthias Ruff, Heinz P. Singer, and Juliane Hollender; Environmental Science &

Technology 2014 48 (4), 2097-2098; DOI:10.1021/es5002105

Response: We thank referee for this very good suggestion. The level of identification confidence for our HRMS dataset and results has been addressed in the revised manuscript, as follows: "Since the measurements of fragmentation patterns and reference standards were not available herein, the identification confidence of the presented results belongs to the unequivocal molecular formula level, according to Schymanski et al. (2014)." (Lines 130–132)

"Schymanski, E. L., Jeon, J., Gulde, R., Fenner, K., Ruff, M., Singer, H. P., and Hollender, J.: Identifying small molecules via high resolution mass spectrometry: Communicating confidence, Environ. Sci. Technol., 48, 2097–2098, https://doi.org/10.1021/es5002105, 2014." (Lines 754–756)

9. L308: You should discuss this more in detail as all three groups CHOS, CHON and CHONS are influenced by anthropogenic emissions. CHOS and CHON don't show any difference between summer and winter - why does the CHONS subgroup (especially in ESI-)?

Related sentence: "In contrast, prominent increases were observed for the number fractions of CHO and CHON species in ESI+ mode and CHONS species in both modes at approximately C6-11 range during the winter period, probably resulted mainly from the enhanced influence of anthropogenic pollution from the surrounding areas, compared with those of the summer period."

Response: As suggested by the referee, we have added more discussions regarding the different variation patterns of individual subgroups in the revised manuscript, as follows:

"Nevertheless, the degrees of influence could vary across individual subgroups, as indicated by the different variation patterns in their carbon atoms number distribution. The larger variations of CHONS species in both modes might result from the stronger influence of anthropogenic pollution compared with those of other species." (Lines 316–319).

**10. L496: Delete during.**

Related sentence: "The molecular composition of organic aerosols in atmospheric PM2.5 was characterized using UHPLC–HRMS at a forest site in the Qinling Mountains region of central China during contrasting summer and winter seasons of 2021/2022."

Response: Since the "contrasting" here is an adjective, it would be not quite reasonable to only delete the "during" in this sentence. Therefore, we have revised the original sentence to make it clearer, as follows:

"The molecular composition of organic aerosols in atmospheric PM2.5 was characterized using UHPLC–HRMS at a forest site in the Qinling Mountains region of central China during **the two contrasting seasons** (summer and winter) of 2021/2022." (Lines 505–506)